# Increasing certainty in systems biology models using Bayesian multimodel inference

Nathaniel Linden-Santangeli [1], Jin Zhang[2], Boris Kramer [1] ✉ &
Padmini Rangamani [1,2] ✉

Mathematical models are indispensable for studying the architecture and behavior of intracellular signaling networks. It is common to develop models using phenomenological approximations due to the difficulty of fully observing the intermediate steps in intracellular signaling pathways. Thus, multiple models can be built to represent the same pathway. This opens up challenges for model selection and decreases certainty in predictions. Here, we investigate Bayesian multimodel inference (MMI) as an approach to increase certainty in systems biology predictions, which becomes relevant when one wants to leverage a set of potentially incomplete models. Using existing models of the extracellular-regulated kinase (ERK) pathway, we show that MMI successfully combines models and yields predictors robust to model set changes and data uncertainties. We then use MMI to identify possible mechanisms of experimentally measured subcellular location-specific ERK activity. This work highlights MMI as a disciplined approach to increasing the certainty of intracellular signaling activity predictions.

Current innovations in molecular tools[1,2] and high-resolution microscopy[3,4] have led to discoveries in intracellular signal transduction, such as the spatial regulation of signaling pathways. These discoveries often require new mathematical models to give rise to mechanistic insights. Furthermore, mathematical models also enable the generation of experimentally testable predictions of intracellular signaling[5–7]. However, one key challenge in systems biology is formulating a model when there are many unknowns, and the available data do not observe every species in a system. As a result, different mathematical models that vary in their simplifying assumptions and model formulations can describe the same signaling pathway. For example, searching the popular BioModels database for models of the *extracellular-regulated kinase* (ERK) signaling cascade yields over 125 results for models that use ordinary differential equations[8,9]. While each model was developed for specific experimental observations, accounting for the range of simplifying assumptions can help improve the certainty and accuracy of resulting predictions. In this work, we explore two important questions: (1) *How can we quantify the effects of uncertainty in the model formulation, called model uncertainty, on model predictions?* and (2) *How can we reduce model choice uncertainty to increase the certainty of intracellular signaling predictions?* Using ERK signaling as our model system, we show that *Bayesian multimodel inference* (MMI) helps to address model uncertainty by leveraging the available data and accounting for all user-specified models.

Uncertainty quantification in systems biology aims to understand how model assumptions, inferred quantities, and data uncertainties impact model predictions[10,11]. Bayesian parameter estimation quantitatively assesses parametric uncertainty by estimating a probability distribution for unknown parameters, such as reaction rate constants and equilibrium coefficients, from training data[10,12,13]. However, at the model level, explicit approaches to handle model uncertainty are yet to be routinely employed in systems biology. Model selection using information criteria, such as the Akaike information criterion (AIC),[14,15] or Bayes Factors[16] has been the preferred approach in systems biology to select a single "best" model when multiple models are available[14,17]. Reasoning with a single set of predictions, traditionally from a single model, is much easier than with predictions from multiple models. However, given that the available experimental data are often sparse

[1]Department of Mechanical and Aerospace Engineering, University of California San Diego, La Jolla, CA, USA. [2]Department of Pharmacology, University of California San Diego, La Jolla, CA, USA. ✉e-mail: bmkramer@ucsd.edu; prangamani@ucsd.edu

and noisy, these approaches may limit predictive performance by introducing selection biases and misrepresenting uncertainty[14,18].

Multimodel inference reduces selection biases and accounts for model uncertainty by including contributions from every specified model[13,14,18–21]. Theoretical results have shown that MMI can improve predictive performance by reducing uncertainty and increasing robustness to modeling assumptions[14,18,22–24]. MMI methods combine model predictions by taking a weighted average over all supplied models[23–25]. Methods for choosing the weights range from parametric consensus estimation[23,24] and frequentist Akaike-weighing[14] to *Bayesian model averaging* (BMA)[19], *pseudo-Bayesian model averaging* (pseudo-BMA)[18,22], and *stacking of predictive densities* (stacking)[22].

Previous applications of MMI to systems biology have focused on a limited subset of the available methods, primarily information-criterion-based and BMA approaches. Stumpf et al. conducted a theoretical analysis of MMI for biological network inference, where the network structure was inferred from data[20,21]. Based on these analyses and limited applications to protein-protein interaction data, the authors concluded that choosing a good set of models is generally important for constructing accurate MMI estimates and that MMI could improve predictions of the biological network structures even when the true networks were not among the set of candidate models. More recently, Beik et al. utilized MMI with BMA to select candidate tumor growth mechanisms consistent with several experimental datasets[26]. However, MMI has not yet been investigated to increase the certainty of intracellular signaling predictions in systems biology.

In this work, we expand the Bayesian systems biology toolkit to employ MMI as a powerful approach to address model uncertainty. We apply MMI to extracellular-regulated kinase signaling (ERK) as an example problem for deeper analysis. Specifically, we select ten ERK signaling models that emphasize the core of the ERK pathway and estimate the kinetic parameters with Bayesian inference using experimental data from Keyes et al.[27]. First, we show that MMI increases the certainty of model predictions using several different experimental and synthetic datasets. Next, we show that MMI-based predictions are robust to changes in the composition of the set of supplied ERK models and to increases in data uncertainty. Finally, we apply MMI to study the mechanisms driving ERK activity, which is localized to subcellular locations observed by Keyes et al.[27]. We refer to this ERK activity as subcellular location-specific ERK activity. Subcellular localization of intracellular signaling is hypothesized to play a role in differentiating cellular responses to heterogeneous stimuli and is observable using modern molecular tools[27]. We find that Bayesian MMI enables us to compare hypotheses about what drives location-specific ERK signaling and, in doing so, suggests that location-specific differences in both Rap1 activation and negative feedback strength are necessary to capture the observed dynamics. We conclude that MMI increases predictive certainty when multiple models of the same signaling pathway are available via a structured approach to handle model uncertainty and selection simultaneously.

## Results

### Bayesian multimodel inference
Bayesian multimodel inference systematically constructs a new consensus estimator of important systems biology *quantities of interest* (QoIs) that accounts for model uncertainty. Here, we consider ODE-based intracellular signaling models, $\mathcal{M}_k$, with fixed model structure and unknown model parameters. We leverage the Bayesian framework to estimate unknown parameters and characterize predictive uncertainty through predictive probability densities[10,13] (see "Methods" for more details). The training data, $d_{\text{train}} = \{\mathbf{y}^1, \ldots, \mathbf{y}^{N_{\text{train}}}\}$, consists of $N_{\text{train}}$ noisy experimental observations (or synthetic data), and can correspond to time points $t^i$ in dynamic responses $\mathbf{y}^i = \mathbf{y}(t^i)$ or to input stimuli $u^i$ in dose-response curves $\mathbf{y}^i = \mathbf{y}(u^i)$. Given that we use Bayesian

methods to estimate unknown model parameters from training data, each model predicts a probability density for the QoI, $\text{p}(\hat{q}_k)$[12,28]. The goal of Bayesian MMI is to build a multimodel estimate of the QoI, $q$, defined as $\text{p}(q|d_{\text{train}}, \mathfrak{M}_K)$, that leverages the entire set of specified models, $\mathfrak{M}_K = \{\mathcal{M}_1, \ldots, \mathcal{M}_K\}$[14,19,22,29]. For intracellular signaling, the QoIs can be either the time-varying trajectories of activities or concentrations of key biochemical species ($q(t)$ at time $t$) or the steady-state dose-response curve ($q(u_i)$ where $u_i$ is a stimulus concentration, and $u_i < u_{i+1}$). Time-varying trajectories characterize the dynamic response of a signaling network to prescribed stimuli. On the other hand, dose-response curves capture the input-output response of the entire signaling network over a range of stimuli concentrations. Measurements of both quantities are fundamental to experimentally characterizing intracellular signaling; for example, see ref. 30 for EGF-ERK dose-response data, and see ref. 27 for time-varying ERK activity data.

The Bayesian multimodel inference workflow used in this work is shown in Fig. 1, where we first calibrate available models to training data by estimating unknown model parameters with Bayesian inference, then we combine the resulting predictive probability densities using MMI, and finally provide improved multimodel predictions of important quantities in systems biology studies. Bayesian multimodel inference constructs predictors by taking a linear combination of predictive densities from each model,

$$\text{p}(q|d_{\text{train}}, \mathfrak{M}_K) := \sum_{k=1}^{K} w_k \text{p}(q_k|\mathcal{M}_k, d_{\text{train}}), \qquad (1)$$

with weights $w_k \geq 0$ and $\sum_k^K w_k = 1$[14,19,25,29,31]. We note that the weights can either be scalars or realizations of a probability mass function defined over the set of models. The key challenge is estimating the weight to assign each predictive density. Here, we compare three methods for choosing the weights, $w_k$: BMA[19], pseudo-BMA[22,29], and stacking[22]. Notably, the potential for MMI to bring new insights into intracellular signaling has not previously been explored, and because each method for MMI has distinct advantages and disadvantages, we chose to compare all three. We briefly summarize the methods below. Importantly, the MMI workflow and methods we investigate in this work broadly apply beyond systems biology whenever a set of candidate models are available.

BMA uses the probability of each model conditioned on the training data to assign weights $w_k^{\text{BMA}} = \text{p}(\mathcal{M}_k|d_{\text{train}})$[19] to each model. The model probability quantifies the probability of model $\mathcal{M}_k$ correctly predicting the training data relative to the other models in the set. While BMA is the natural Bayesian approach to MMI, the method suffers from several key challenges, including the necessary computation of the marginal likelihood, strong dependence on prior information, and reliance on data-fit alone instead of on predictive performance[13,19,22,32]. Furthermore, as with all fully Bayesian methods, the resulting BMA weights vary with the amount of training data provided, and in the limit of infinite data, the weights converge to a single model[22]. Due to these potential challenges of BMA, we also investigate pseudo-BMA and stacking.

Pseudo-Bayesian model averaging assigns model weights based on the expected predictive performance of unseen data measured with the *expected log pointwise predictive density* (ELPD)[18,22]. The ELPD quantifies expected predictive performance on new data by computing the distance between the predictive and true data-generating densities and is defined as

$$\text{ELPD}_{\mathcal{M}_k} := \sum_{i=1}^{N_{\text{train}}} \int \log \text{p}(\tilde{\mathbf{y}}_i|d_{\text{train}}, \mathcal{M}_k) \text{p}_{\text{true}}(\tilde{\mathbf{y}}_i) \, d\tilde{\mathbf{y}}_i, \qquad (2)$$

where $\mathcal{M}_k$ is the $k$th model and $\text{p}_{\text{true}}(\tilde{\mathbf{y}}_i)$ true data-generating distribution $\text{p}_{\text{true}}(\tilde{\mathbf{y}}_i)$. Higher ELPD values indicate a greater probability

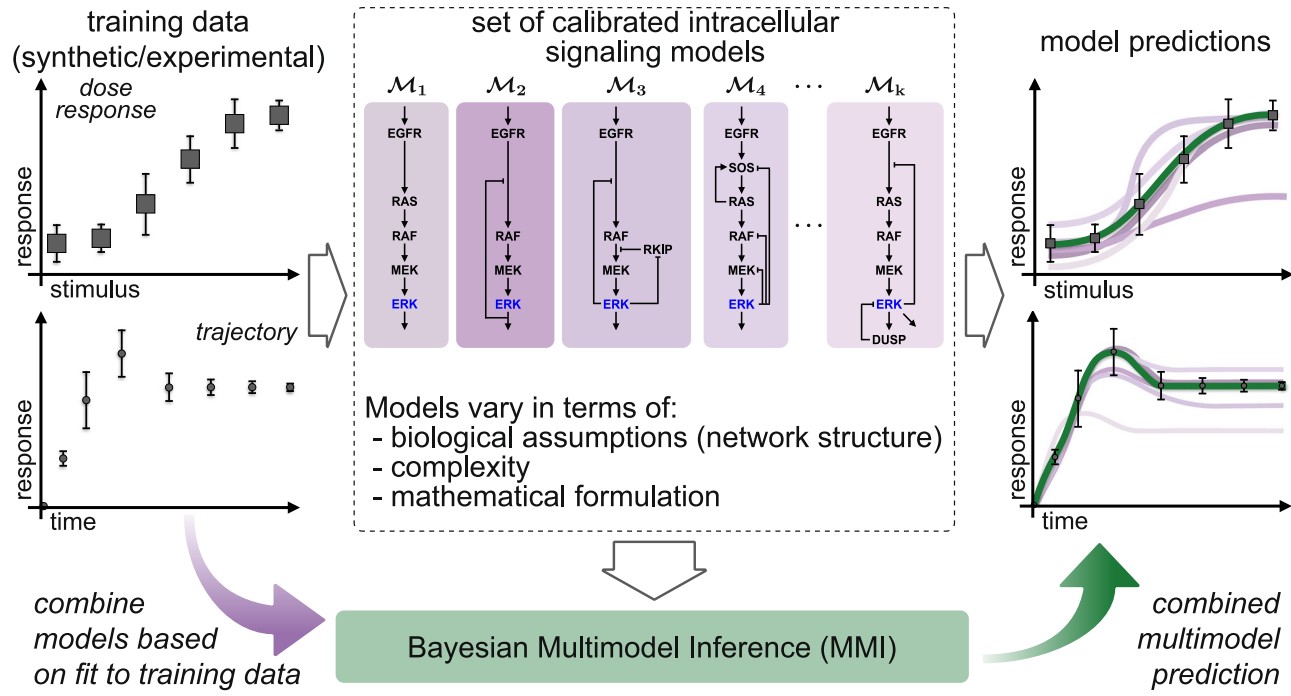

**Fig. 1 | Bayesian multimodel inference (MMI) accounts for model uncertainty by incorporating contributions from intracellular signaling models specified by the user.** First, we use Bayesian parameter estimation to calibrate the parameters in each model to the training data. Each calibrated model can then be used to predict biological quantities; however, the accuracy and uncertainty of each model's predictions may vary due to differences in the model structure and mathematical formulation. Next, Bayesian MMI constructs a new estimator by combining specified models based on how well each predicts the training data or based on the expected predictive performance of future data. Finally, the multi-model inference-based estimator improves predictive accuracy and robustness compared to individual models. In this work, we use models of extracellular-regulated kinase signaling (ERK) to apply Bayesian MMI for systems biology. The models vary in mathematical and biological complexity. The Bayesian MMI work-flow is broadly applicable whenever multiple models are available.

of correctly predicting unseen data and, therefore, better predictive performance. Notably, the ELPD is a direct estimate of the predictive performance of a model and does not suffer from the same challenges that affect the marginal likelihood in BMA[18,22,33]. However, the ELPD is intractable directly because we do not know the true data-generating density. Instead, we estimate the ELPD with Pareto smoothed importance sampling leave-one-out cross-validation (PSIS-LOO-CV)[18]. Pseudo-BMA constructs model weights, $w_k^{pBMA}$, by normalizing the estimated ELPD of each model, $\widehat{ELPD}_k^{LOO}$, to the sum of that quantity across all models. The two key challenges of pseudo-BMA are that it relies on potentially erroneous estimates of the ELPD, such as PSIS-LOO-CV, and that pseudo-BMA weights do not account for correlations between individual model predictions[22].

Stacking of predictive densities selects optimal model weights, $w_k^{stack}$, to maximize the ELPD of the consensus density. We follow the approach introduced in ref. 22, which maximizes the log score between the consensus predictive density and the true data-generating distribution estimated by the ELPD. In contrast to BMA and pseudo-BMA, which weigh models independently, stacking weighs all models simultaneously[31]. Thus, stacking finds the best-estimating density that is closest (in terms of the log scoring rule used to define the optimality criterion) to the data-generating process[22]. However, similar to pseudo-BMA, stacking relies on ELPD estimates for MMI.

**Variations in ERK signaling models show model form uncertainty**

ERK signaling plays a key role in controlling many cellular processes, including proliferation, growth, metabolism, and differentiation[34,35]. Due to this widespread importance, ERK is one of the most extensively modeled intracellular signaling pathways[36]. As a test problem for MMI, we specified a set of ten models of the core ERK signaling

network[30,37–45]. Throughout this work, we refer to each model by the first letter of the first author's last name and the publication year, e.g., Huang and Ferrell 1996[30] is H' 1996. Even though each model was initially used to answer different research questions, we chose these ten because each one represents the ERK signaling pathway from input to ERK activation. Fig. 2 depicts the epidermal growth factor (EGF)-to-ERK signaling network spanned by each model and denotes which model includes specific structural components. We note that in addition to each model containing a subset of the complete ERK signaling network, the models also vary in mechanistic complexity and the mathematical formulation used to represent the reaction kinetics. Examples of these variations include using only mass action kinetics in H' 1996 while K' 2017 combines Michaelis-Menten and Hill-equations. Additionally, all models use single equation representations of EGF-EGFR interactions, except for H' 2005 and B' 2007, which use detailed multi-step ligand-binding and receptor internalization mechanisms. Supplementary Table 1 highlights how extensive variations in the number of model parameters and state variables reflect the variations in the model formulation. Furthermore, each model's equations were readily available from original or secondary sources. Therefore, these models exemplify real-world uncertainties while remaining comparable in their representation of the ERK signaling pathway. Thus, we reasoned that this set of ERK models serves as a good test case for MMI.

We previously showed that a priori structural identifiability and global sensitivity analyses are critical to successful Bayesian parameter estimation for intracellular signaling models[12]. Therefore, we performed local structural identifiability analysis on each model to determine which parameters can be uniquely identified locally in parameter space. Next, we performed global sensitivity analysis on each model using Morris screening[46] to determine which identifiable parameters significantly influence predicted ERK activity

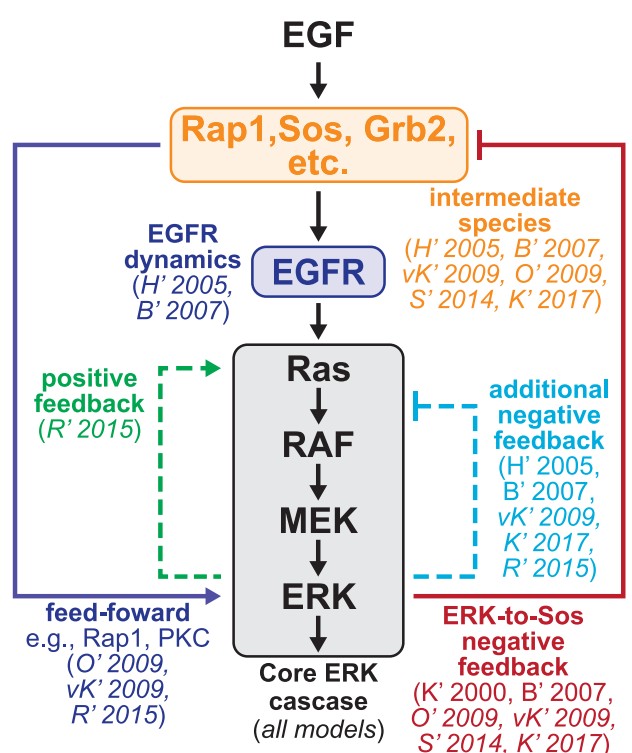

**Fig. 2 | Set of ERK signaling models highlights model form uncertainty due to differences in biological assumptions, model complexity, and mathematical formulation.** Parentheses list the models that include a representation of each component. Abbreviations for models show the first letter of the first author's last name and the publication year, e.g., H' 1996 corresponds to Huang and Ferrell 1996[30]. Supplementary Table 1 lists the abbreviations and references for all models.

(Supplementary Fig. 1). Based on the results of both analyses, we reduced the number of free parameters by fixing all nonidentifiable and noninfluential parameters to nominal values from the literature. Supplementary Table 1 shows the number of parameters remaining after each of these a priori analyses. We only estimated the remaining free parameters.

### Bayesian MMI increases certainty in ERK activity predictions

As a first test, we applied MMI to predict EGF-induced ERK activity from experimental observations of cytoplasmic ERK kinase activity from Keyes et al.[27]. Here, we sought to understand better how model averaging with MMI impacts predictions of time-varying EGF-induced ERK activity. Using an improved ERK kinase activity reporter called EKAR4, Keyes et al. measured ERK activity at subcellular locations, including the cytoplasm (Fig. 3a) and plasma membrane (Supplementary Fig. 3a). Using the cytoplasmic ERK activity, we separately estimated the parameters of the ten ERK signaling models. Figure 3b shows the resulting posterior densities for the ERK activity predictions with four of the ten models (Supplementary Fig. 2a shows the densities for all ten models). All ten models predict the cytoplasmic ERK activity data qualitatively well. We attribute minor variations in the predictions to differences in the formulation of each model.

Each of the three MMI methods distributed MMI weights differently between the models, but the MMI predictions are similar. The ranking of the models using the ELPD values (Fig. 3c) and the model probabilities (Fig. 3d) varies between the two methods, despite the overall trends remaining similar. We observe that the models that predict cytoplasmic ERK activity qualitatively better had higher ELPD values and higher model probability than those that did poorly. Both pseudo-BMA and BMA assigned non-zero weights to more than one model, but they weighed different subsets of the model set (Fig. 3e).

Meanwhile, stacking placed all weight on the O' 2009 model. Despite the different weight distributions, each of the three MMI methods predicted trajectories of cytoplasmic ERK activity that all appear qualitatively similar to the data (Supplementary Fig. 2b). Interestingly, the models with higher ELPD values were not necessarily the most complex or the models with the most free parameters. The best models, O' 2009 and K' 2017, had a moderate number of states and included components such as intermediate species and negative feedback (see above). This implies that there is minimal mechanistic complexity which is required to capture time-varying cytoplasmic ERK activity.

Quantitative differences in the predictions for each method show that only pseudo-BMA and BMA increased predictive certainty. The *root mean square error* (RMSE) of the posterior mean and the predictive uncertainty measured by the average width of the posterior 95% credible interval are slightly lower than that of the best model for pseudo-BMA and BMA (Fig. 3f, g). However, these quantities are the same as the best model for stacking, which only weighs a single model. These results show that MMI predictions of cytoplasmic ERK activity retain the predictive accuracy of the best models and can also yield reductions in uncertainty when a weighted average is used.

To test if MMI improves predictions for additional ERK signaling data, we repeated MMI for the same set of models using experimental measurements of plasma membrane ERK activity (Supplementary Fig. 3a) and synthetic EGF-ERK dose-response curve data (Supplementary Fig. 4a). For both of these new datasets, predictions varied between individual models and thus, the ELPD estimates and model probabilities varied as well (Supplementary Figs. 3b–d and 4b–d). Accordingly, each MMI method yields different weight assignments (Supplementary Figs. 3e and 4e). We note that several models in the dose-response example, e.g., O' 2009, failed to capture the trends in the dose-response data (Supplementary Fig. 4b). Importantly, these models received nearly no weight in the MMI averages and do not contribute to MMI predictions. For both datasets, MMI predictions had predictive errors on par with the best individual models. Additionally, MMI predictions had uncertainty that was lower than that of individual models when multiple models were combined via weighted averaging (Supplementary Figs. 3f, g and 4f, g). Therefore, we concluded that MMI handles model uncertainty in the set of specified models for predictions of plasma membrane ERK activity and the EGF-ERK dose-response curve in addition to cytoplasmic ERK activity. Importantly, from all three datasets, MMI predictions had lower errors than choosing a model at random and showed reduced uncertainty when using pseudo-BMA and BMA. Thus, MMI can increase predictive certainty for this set of models across multiple types of data.

### MMI increases robustness to model form uncertainty

To understand how increasing model uncertainty affects model-averaged MMI predictions, we systematically varied the composition of the model set. Specifically, we asked how (i) adding an ill-fitting model, (ii) removing the best model, and (iii) changing the size of the model set affects MMI predictive performance.

First, we found that single-model perturbations to the model set tended to have minor effects on the performance of MMI predictions. To test this for predictions of cytoplasmic ERK activity, we first excluded the worst-fit model, R' 2015 (lowest ELPD in Fig. 3c), and computed the MMI weights (Fig. 4a). Adding the *bad* model back to the set resulted in little change to the MMI weights (Fig. 4b), and thus, had minimal impact on the predictive error (Fig. 4d) and the posterior uncertainty (Fig. 4e) because the additional *bad* model received nearly no weight. Next, we removed the best-fit model, O' 2009, (highest ELPD), and recomputed the MMI weights (Fig. 4c). Without the *best* model, stacking reassigned all weight to K' 2017, the next best model, while BMA redistributed the weight across K' 2017 and B' 2007. Pseudo-BMA redistributed weight away from O' 2009 across several

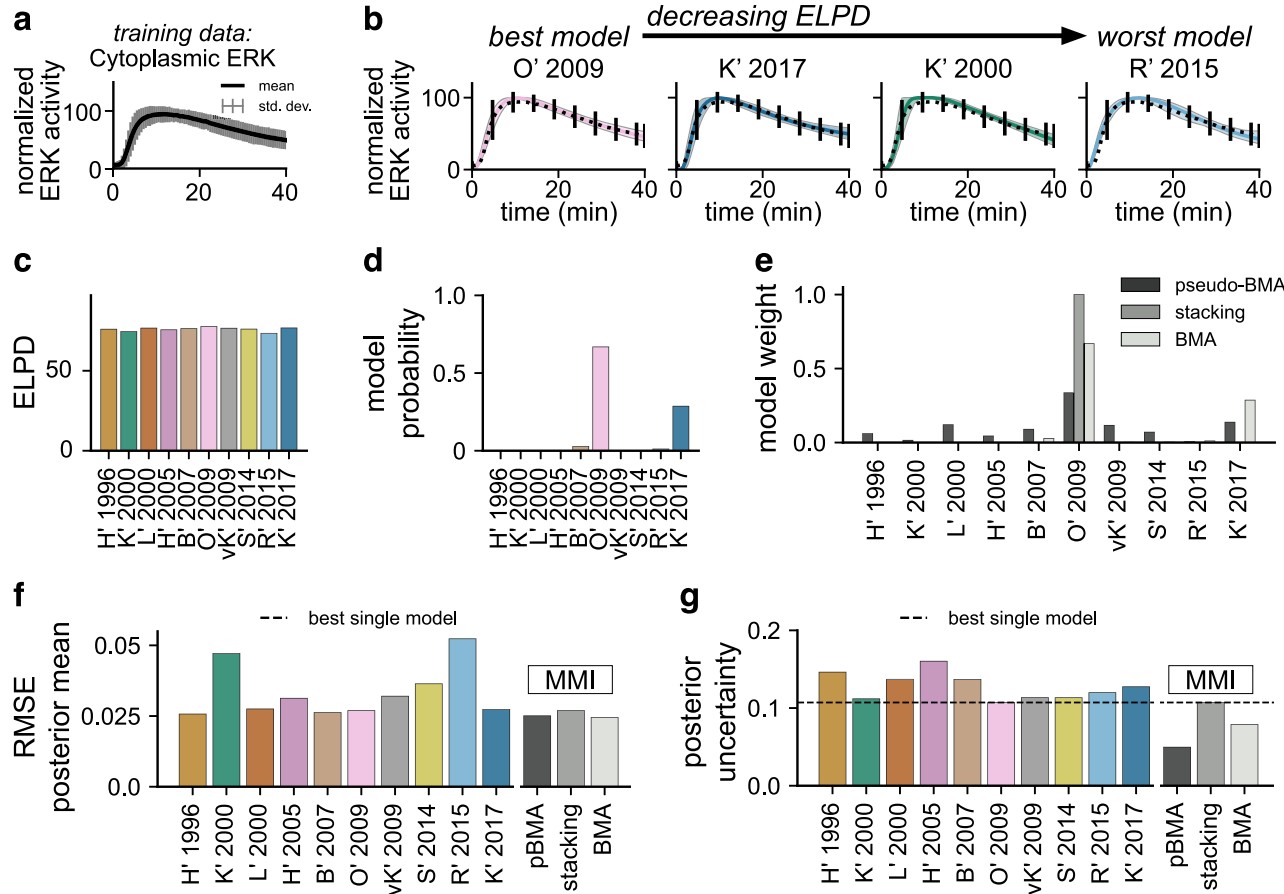

**Fig. 3 | Multimodel inference reduces uncertainty in cytoplasmic ERK activity predictions. a** Normalized experimental measurements of cytoplasmic ERK activity. Mean (black trace) and standard deviation (grey bars) show statistics computed across all single-cell trajectories. Data originally presented in figure 1 of ref. 27. **b** Posterior densities of cytoplasmic ERK activity trajectories for four out of nine models ordered by decreasing expected log pointwise predictive density (ELPD). Additional densities are shown in Supplementary Fig. 2a. Shaded regions show the 95% credible interval (the interval between 2. 5th and 97. 5th percentiles) of the posterior, while the solid line shows the posterior mean. Densities are estimated from $n = 2000$ unique posterior samples. **c** ELPD values for all models computed using Pareto smoothed importance sampling leave-one-out cross-validation (PSIS-LOO-CV). Models with higher relative ELPD are more likely to

predict future data correctly. Estimated ELPD values are used to construct MMI estimates with pseudo-Bayesian model averaging (pseudo-BMA) and stacking of predictive densities (stacking). **d** Model probabilities are computed using sequential Monte Carlo estimates of the marginal likelihood. The model probability is the probability of each model conditioned on the training data and is used for MMI with Bayesian model averaging (BMA). **e** MMI model weights for all models using pseudo-BMA, BMA, and stacking. **f** Root mean square error (RMSE) of the posterior mean cytoplasmic ERK activity prediction. The dashed black line shows the lowest RMSE of any single model. **g** Posterior uncertainty measured by the mean 95% credible interval width in time. The dashed black line shows the lowest uncertainty of any single model.

other models. Accordingly, the predictive error increased only a small amount because all models predict cytoplasmic ERK activity similarly well. Despite the small increases in error, removing the model with the highest ELPD model had little effect on MMI predictive uncertainty (Fig. 4e). We repeated MMI with these single-model perturbations to the model set using both the plasma membrane data (Supplementary Fig. 5a–e, and the synthetic EGF-ERK dose-response data (Supplementary Fig. 6a–e). We observed similar results for both additional datasets; however, the predictive error increased when we removed the best model for the dose-response example because the difference in the performance of the best and next-best models was bigger for that example. Therefore, these results show that MMI predictions of ERK activity are robust to changes in this model set because including an additional *bad* model had little effect, and excluding the *best* model led to relatively small changes in uncertainty and accuracy.

Next, we investigated how many models are necessary for effective MMI and how the number of models in the model set affects MMI predictive performance. We constructed model sets with all possible combinations of the nine ERK models ranging from sets of two to sets

of eight models for all three datasets, cytoplasmic ERK activity (Fig. 4f), plasma membrane ERK activity (Supplementary Fig. 5f) and EGF-ERK dose-response data (Supplementary Fig. 6f). First, we found that the error of MMI predictions on average decreases and approaches the lowest error of any individual model as the number of models increases for all three datasets. In particular, the errors of pseudo-BMA and stacking predictions tended to decrease faster, while the errors of BMA predictions remained elevated even with larger model set sizes. In particular, for BMA predictions of the EGF-ERK dose-response curve, the relative error remains centered around 5%, which is well above the error of the best individual model (Supplementary Fig. 6f). Based on these findings, we conclude that MMI can increase the probability of correct predictions beyond single models, even with just two specified models. However, the improvements were much greater with pseudo-BMA and stacking than with BMA.

How does the number of models affect MMI predictive uncertainty? For all three datasets, the predictive uncertainty decreased as model set size was increased for pseudo-BMA and BMA, but not for stacking (cytoplasmic ERK activity, Fig. 4g; plasma membrane ERK

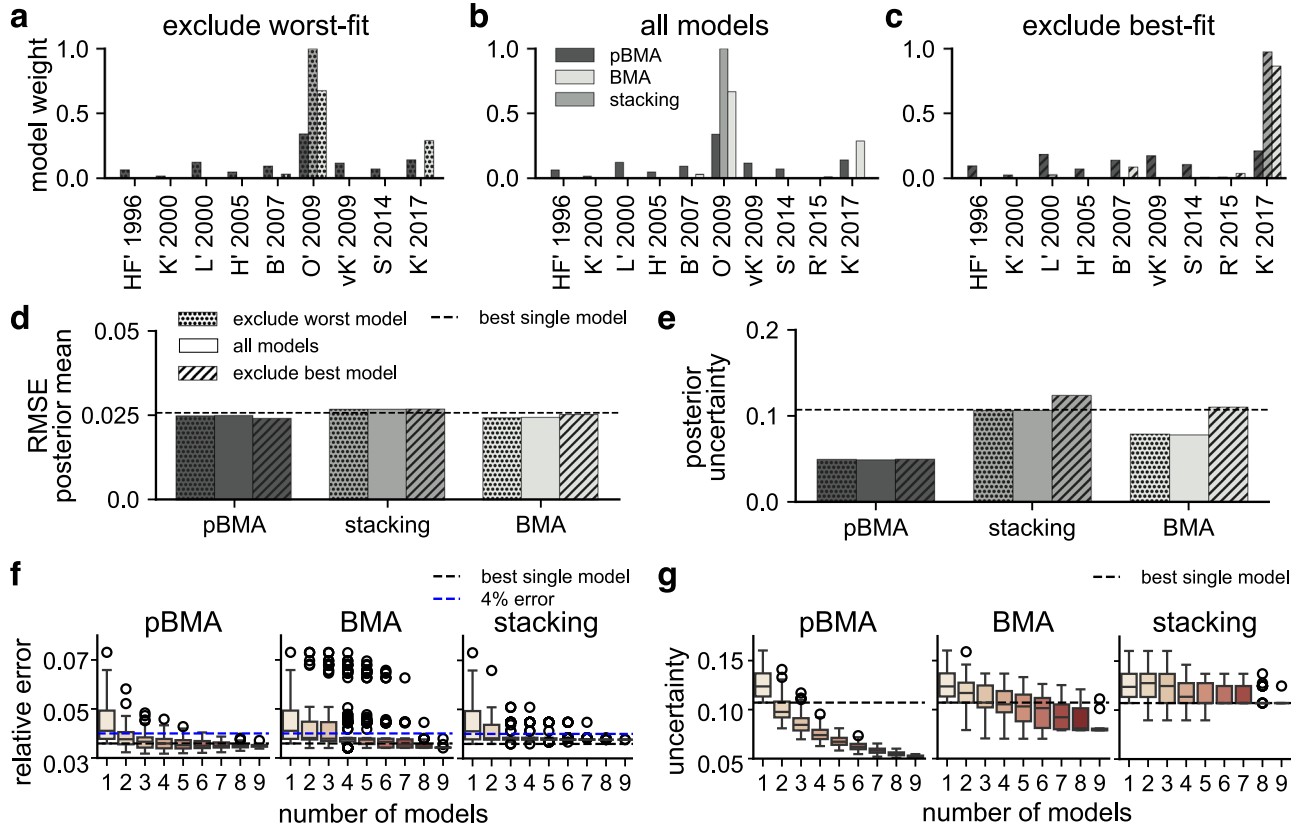

**Fig. 4 | Bayesian multimodel inference is robust to perturbations in the set of plausible models for cytoplasmic ERK activity trajectory predictions.** **a–c** Weights assigned to models in three model sets: **a** excluding the worst-fit model, R' 2015, **b** all models, **c** excluding the best-fit, O' 2009. **d** RMSE of the posterior mean cytoplasmic ERK activity prediction for each MMI method and model set. Dotted patterning corresponds to (**a**), no patterning to (**b**), and dashed patterning to (**c**). The dashed horizontal line is the RMSE of the model with the lowest average error, HF' 1996. **e** Posterior uncertainty is measured by the mean of the 95% credible interval taken over all EGF levels. **f** Relative error of the posterior

mean for MMI predictions with increasing model set size. The dashed blue line shows 5% relative error (0.05), and the dashed black line shows the lowest error of any single model, HF' 1996. **g** Average posterior uncertainty of ERK response for MMI predictions with increasing numbers of models. The dashed black line shows the uncertainty of the best model, R' 2015. **f**, **g** Boxplots show the mean and interquartile range, and whiskers show the 95% credible interval. Open circles show outliers. All possible combinations of models were tested at each size, for example, for set of two models, we included $n = 45$ (ten choose two) unique combinations.

activity, Supplementary Fig. 5g; EGF-ERK dose-response Supplementary Fig. 6g). For pseudo-BMA and BMA, the uncertainty approached a level that is below that of the best model (i.e., the dashed black line in Fig. 4g) as the number of models increased. However, for stacking, the uncertainty is similar to the prediction made with the model with the greatest ELPD, even after more than three models are used. Therefore, we conclude that MMI reduces uncertainty even with a moderate number of models and that more models lead to greater uncertainty reductions for pseudo-BMA and BMA in these three examples. Additionally, these results show that even with the increased model uncertainty from having more models, MMI can reliably increase predictive certainty and generate predictions as accurately as the best single models.

## MMI is robust to increased training data uncertainty

We have previously observed that data uncertainty can greatly impact predictive uncertainty in intracellular signaling models[12]. Furthermore, the quantity and quality of training data affect how different MMI methods weigh models[22]. To explore how MMI performs in the presence of data uncertainties, we simulated increasing data uncertainty by decreasing the quantity and quality of the training data supplied for MMI.

First, we found that MMI predictions of future cytoplasmic ERK activity have lower uncertainty than single-model predictions. We varied the length of the training data by truncating the original 40-min

long recordings at the 10-, 20-, and 30-min time points (Fig. 5a), and used MMI to predict the remainder of the trajectory up to 40 min. We evaluated the predictions by assessing the testing error and uncertainty on the final 10 min of the trajectory, which was not used for parameter estimation. Similar to all individual models, the posterior uncertainty for the best single model, O' 2009, increased with decreasing training data length (top row in Fig. 5b; additional individual model predictions are shown in Supplementary Fig. 7). However, the pseudo-BMA predictions showed substantially lower predictive uncertainty at all data lengths (bottom row in Fig. 5b). Quantitatively, the testing error of MMI predictions is on par with that of the lowest-error models at all training data lengths (Fig. 5c), and the testing uncertainty of MMI predictions is substantially lower than the best single models (Fig. 5d). We note that at the 10- and 20-min data lengths, the L' 2000, K' 2000, and H' 2005 predictions have lower uncertainty than those for MMI. However, they are highly erroneous, with almost all simulations predicting maximal activation after 40 min (Supplementary Fig. 7). We additionally repeated the same simulations with data from the plasma membrane. We observed similar trends as with cytoplasm data (Supplementary Fig. 8a–e). Together, these results show that MMI increases the certainty and accuracy of out-of-sample predictions of future ERK activity from shorter training data than single models alone.

Second, MMI predictions showed lower predictive error and uncertainty than single-model predictions from lower-quality data. We

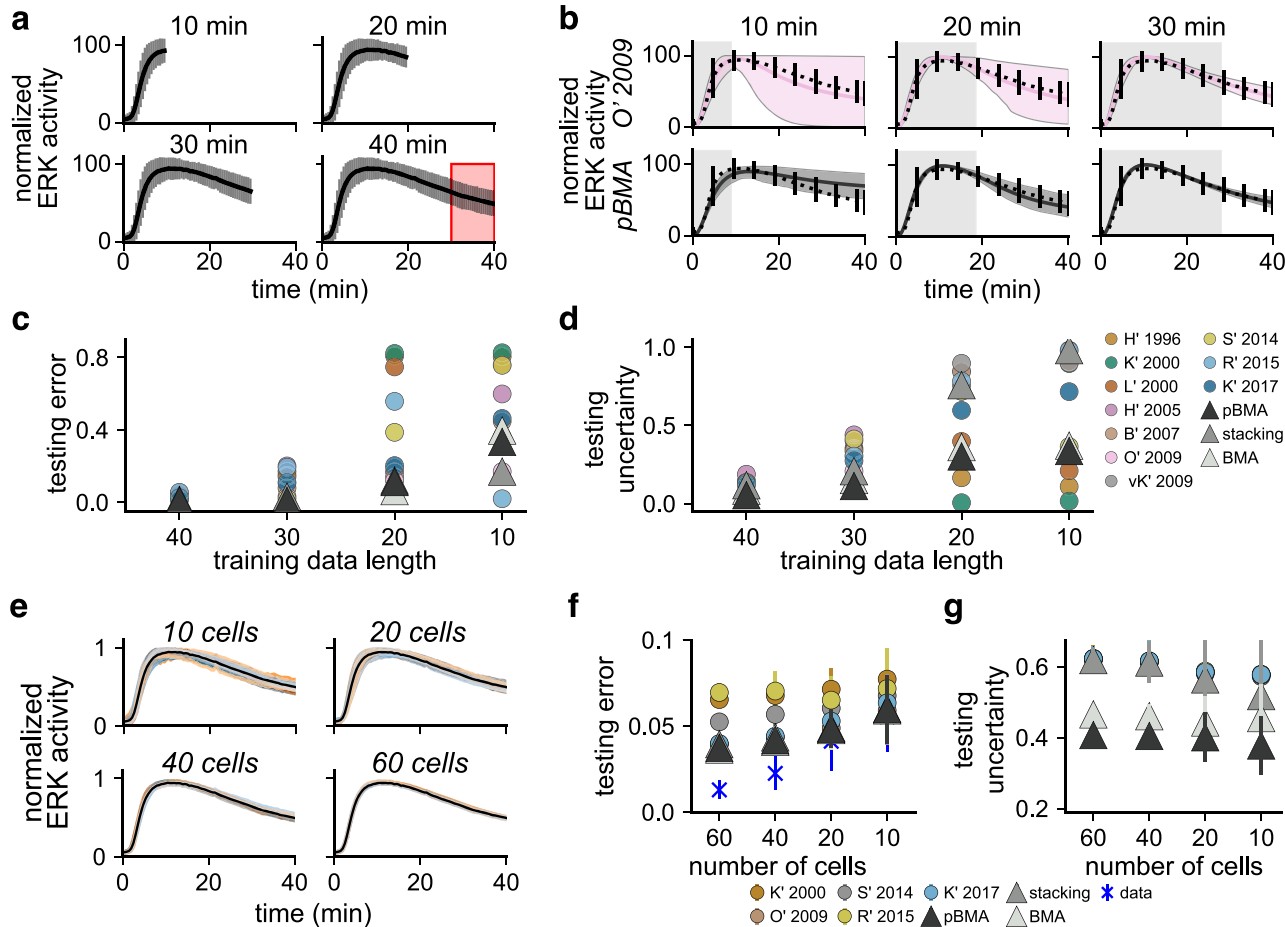

**Fig. 5 | Bayesian mulitmodel inference predictions of cytoplasmic ERK activity are robust to uncertainties due to decreasing data length and quality. a** Shorter training data was constructed by truncating the original 40-min cytoplasmic ERK trajectories at the 10-, 20-, and 30-min time points (mean, black trace; and standard deviation, grey bars). Predictive performance was assessed by computing errors and uncertainties in the final 10 min (red box). **b** Posterior predictions from decreased training data for the best model, O' 2009 (highest ELPD across all training datasets), and MMI with pseudo-BMA. Error bars show the 95% posterior credible interval using $n = 500$ unique posterior samples. Predictions for additional models and MMI methods are shown in Supplementary Fig. 7. **c** Predictive error (relative error) for the final 10 min ($t = 30 \rightarrow t = 40$ min) of cytoplasmic ERK activity.

**d** Predictive uncertainty (average width of the 95% credible interval) for the final 10 min of cytoplasmic ERK activity. **e** Lower-quality training data was generated by averaging random subsets of 10, 20, 40, and 60 imaged cells. The black trace shows an original average of 76 cells. Colored traces show averages of 40 replicate random subsets. **f** The predictive error (relative error) of cytoplasmic ERK activity with lower quality data compared to average activity trajectory using all cells. **g** Predictive uncertainty of cytoplasmic ERK activity with lower quality data. **f, g** Filled circles indicate an average error of 40 replicates for individual models and triangles for MMI predictions. Error bars show the standard deviation over replicates. Blue markers show the error of the raw training data at each subset size compared to the original full-data mean.

generated lower-quality training data by decreasing the number of individual cells we averaged over to create the normalized data. Specifically, we drew 40 random subsets of 10, 20, 40, and 60 of the 76 original single-cell recordings of cytoplasmic ERK activity and computed the cell-wise average and standard deviation (Fig. 5f). Data created with lower cell counts (10 or 20 cells) was of lower quality than data with higher counts (40 or 60 cells) because the lower-count data had greater error compared to the original 76-cell dataset and had more variation in the uncertainty (Supplementary Fig. 9). To reduce the computational burden of repeated parameter estimation, we constructed MMI estimates using five of the ten models, K' 2000, O' 2009, S' 2014, R' 2015, and K' 2017, because these models had the shortest inference times (Supplementary Table 3). The predictive testing error, computed as the relative error of model predictions compared to the original 76-cell dataset, shows that MMI predictions are more accurate than most of the individual models (Fig. 5g). Additionally, the BMA and pseudo-BMA MMI predictions had lower predictive uncertainty across all subset sizes (Fig. 5h). We repeated the same simulations with data from the plasma membrane. Similarly, we found that MMI is robust to lower-quality data

(Supplementary Fig. 8f, g). Based on these results, we conclude that MMI predictions are more robust to uncertainties due to reduced data quality than single-model predictions.

**Subcellular ERK activity depends on Rap1 and negative feedback**

Keyes et al. previously revealed spatiotemporal differences between cytoplasmic and plasma membrane ERK signaling[27]. Specifically, the authors observed that ERK activity is more sustained at the plasma membrane compared to the cytoplasm (Fig. 6a, black traces; data reproduced from figures 1C and 1D of ref. 27). Furthermore, Keyes et al. showed that plasma membrane ERK activity depended strongly on the non-canonical ERK activator Rap1, but cytoplasmic activity did not (figures 3C and 3D from ref. 27). Here, we used MMI to investigate whether subcellular differences in the model parameters could predict subcellular variations in ERK activity by selecting models that accurately explain the data.

Initially, we hypothesized that location-specific probability densities for model parameters could explain location-specific ERK activity observed by Keyes et al.[27]. In the previous sections, we observed that the same models could predict both cytoplasmic and plasma

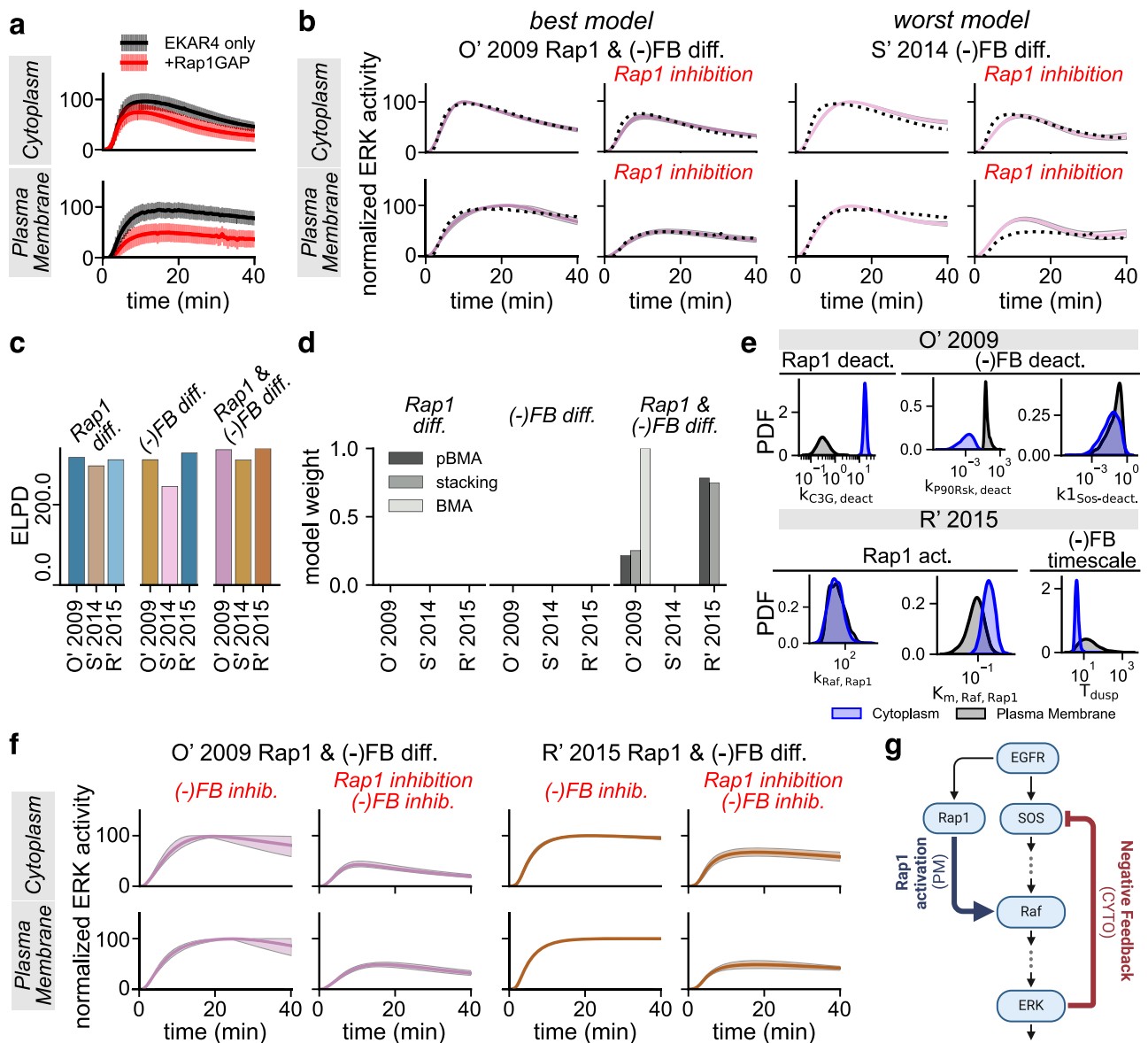

**Fig. 6 | Bayesian mulitmodel inference suggests that subcellular differences in ERK activity depend on both Rap1 activity and ERK negative feedback. a** EKAR4 measurements from ref. 27 Figure 3C, D normalized to the average maximum EKAR4 emission ratio in each location separately. Mean and standard deviation for EKAR4-only (black) and EKAR4 with Rap1 inhibition measurements (+Rap1GAP; red). **b** Posterior predictions for the best and worst models indicated by ELPD, i.e., O' 2009 with location differences in Rap1 and negative feedback (O' 2009 Rap1 & (−)FB diff.) and S' 2014 with location differences in negative feedback (S' 2014 (−)FB diff.), respectively. Predictions from all models are shown in Supplementary Fig. 11. Error bars show the 95% posterior credible interval. **c** ELPD values for all models. **d** MMI weights assigned to all models using pseudo-BMA, stacking, and BMA. **e** Marginal posterior densities for the Rap1 activation/deactivation and ERK

negative feedback parameters in each location for the O' 2009 and R' 2015 models with Rap1 and negative feedback location differences selected by MMI. Cytoplasm densities (CYTO) are blue, and those for the plasma membrane (PM) are black. All densities are statistically independent between locations ($p < 0.005$ by the Kolmogorov–Smirnov test with a two-sided hypothesis). **f** Posterior predictions for selected models with negative feedback inhibition alone ((−)FB inhib.) and simultaneous Rap1 and negative feedback inhibition (Rap1 inhibition + (−)FB inhib.). Error bars show the 95% posterior credible interval. **g** The proposed mechanism of subcellular differences in ERK activity discovered using MMI. Rap1 activation is stronger at the plasma membrane, while negative feedback is stronger in the cytoplasm.

membrane ERK activity well when all parameters can vary independently between the two compartments (see above). Supplementary Fig. 10 highlights each model's most significant subcellular-specific parameter estimates. For example, there were subcellular differences in Raf deactivation, EGFR dynamics, ERK activation, and total kinase concentrations. However, since we allowed all model parameters to vary independently between compartments, these findings show which subcellular parameter differences can lead to more accurate predictions of subcellular ERK activity, but do not show which specific

combinations of independent parameters are required to capture the data.

To explore which combinations of location-specific parameters could capture location-specific ERK activity, we used Bayesian MMI to evaluate all possible combinations of parameters varying between compartments. Specifically, we focused on the O' 2009, S' 2014, and R' 2015 models because these three models comprehensively cover the core ERK network. Furthermore, limiting the investigation to three models ensured the computations remained tractable. In these three

models, we observed that the parameters that control Raf deactivation, ERK phosphorylation, ERK dephosphorylation, ERK negative feedback, and Rap1 activity varied significantly between compartments for these three models (Supplementary Fig. 10 and Supplementary Materials). Therefore, we allowed combinations of the related parameters to vary between compartments. Since the Rap1-dependence of plasma membrane ERK activity was a key finding of Keyes et al. [27], we introduced a Rap1-like activator into the S' 2014 and R' 2015 models, such that all three models could be used to predict ERK activity after Rap1 inhibition. To do so, we assumed that active EGFR activates Rap1, which in turn activates downstream Raf kinase, and that Rap1 activation is not modulated by negative feedback from active ERK[47]. We then performed 94 sets of parameter estimation, allowing every combination of the relevant parameters to either vary independently between compartments or remain the same globally. We fit the models under each hypothesis to EKAR4-only and EKAR4 with Rap1 inhibitor (Rap1GAP) data (Fig. 6a). To simulate Rap1 inhibition, we set the total Rap1 concentration to zero.

Next, we used MMI to compare which of the combinations of location-specific parameter estimation best captured the subcellular ERK activity data. MMI weighted various parameter combinations (Supplementary Table 4 shows the best and worst ten). Eight parameter combinations received weights greater than 0.01. These all similarly allowed ERK phosphorylation and/or ERK dephosphorylation parameters to vary between compartments. Additional combinations of location-specific parameters included Rap1 activation, ERK (−) feedback, and Raf deactivation (Supplementary Table 4). The identified location-specific parameters controlling ERK phosphorylation can indicate species not included in the models are responsible for location-specific dynamics, such as potential cross-talk or other feedback mechanisms. However, it is also possible that these parameters are simply the *easiest* to estimate. Thus, the location-specific differences may not represent biologically realistic mechanisms because of the known ability of systems biology models to fit a wide range of data[48].

To further refine our understanding of location-specific parameters that can capture the experimental data, we used MMI to compare a smaller set of hypotheses about which parameters impact location-specific ERK activity based on what is known to be biologically realistic[27,47,49]. Specifically, we focused on the noncanonical ERK activator Rap1 and ERK negative feedback. We hypothesized that Rap1 signaling would vary between compartments because Keyes et al. [27] found that plasma membrane ERK activity depends on Rap1 while cytoplasmic activity does not (Figure 3 from ref. 27). Additionally, we hypothesized that ERK negative feedback could vary between compartments because previous findings suggest that the strength of ERK negative feedback can drive differences in sustained versus transient ERK activity[47,49]. To test these hypotheses about location-specific signaling, we performed nine sets of parameter estimation in which we allowed only the Rap1 parameters (Rap1 diff.), only the ERK negative feedback parameters ((−)FB diff.), or both Rap1 and ERK negative feedback parameters (Rap1 & (−)FB diff.) to vary independently between locations. Parameters not associated with Rap1 or ERK negative feedback were assumed to not vary between locations, and a single probability density was estimated for those parameters.

Location-specific Rap1 signaling and ERK negative feedback were both necessary to predict ERK activity's subcellular variability. Qualitatively, the posterior predictive distributions of the models that allowed only ERK negative feedback parameters to vary between locations ((−)FB diff.) predict the data worse, while those that allowed Rap1 parameters to vary (Rap1 diff.) predict the data more accurately. The models that allow both Rap1 and ERK negative feedback parameters to vary, Rap1 & (−)FB diff., appeared to predict the data best (Fig. 6c and Supplementary Fig. 11). The predicted ELPDs show that Rap1 & (−)FB diff. models do indeed predict the data best, followed by

Rap1 diff. and (−)FB diff., respectively (Fig. 6c). Furthermore, all three MMI methods only assign weight to hypotheses that allow both Rap1 and negative feedback differences simultaneously (Fig. 6d). Interestingly, pseudo-BMA and stacking select both the O' 2009- and R' 2015-based models, while BMA selects the O' 2009-based model. This suggests that location-specific ERK negative feedback plays a role in location-specific ERK activity in addition to the Rap1 dependence at the plasma membrane.

Interestingly, in the hypotheses selected by MMI, Rap1 activation is stronger at the plasma membrane, and ERK negative feedback is stronger in the cytoplasm as indicated by the marginal posterior densities of the associated parameters (Fig. 6e). Specifically, the deactivation rate of the Rap1 activator C3G ($k1_{\text{C3G-deact.}}$) is lower in the plasma membrane for the O' 2009 model, and the Rap1–Raf equilibrium constant activation rate ($K_{\text{m,Raf,Rap1}}$) is lower at the plasma membrane for the R' 2015 model. Additionally, the parameter estimates suggest stronger ERK negative feedback in the cytoplasm than in the plasma membrane. These results led us to conclude that stronger Rap1 activity drives sustained plasma membrane ERK activity, while stronger negative feedback is necessary for transient cytoplasmic ERK activity. Indeed, inhibition of ERK negative feedback nearly eliminates location-specific differences in ERK activity (Fig. 6f). Inhibiting ERK negative feedback at the cytoplasm leads to ERK activation for both models and, therefore, to the loss of location-specific signaling. However, at the plasma membrane, negative feedback inhibition had little effect on ERK activity. Thus, these results show that ERK negative feedback and Rap1 activity are necessary for location-specific ERK activity.

Here, MMI allowed us to select hypotheses that *best* agreed with the data from a set of preselected, biologically relevant hypotheses. These findings suggest that cytoplasmic ERK activity depends on ERK negative feedback, while plasma membrane ERK activity depends on Rap1. Based on the MMI-selected hypothesis, we propose a new ERK negative feedback- and Rap1-dependent model for subcellular variability in ERK activity (Fig. 6f).

## Discussion

In this study, we demonstrate a novel application of Bayesian multimodel inference to the ERK signaling pathway. MMI enabled us to account for model form uncertainty in systems biology, specifically when multiple models of the same system were available. We compared three methods for MMI—BMA, pseudo-BMA, and stacking of predictive densities. Using ten ERK signaling models, we applied Bayesian MMI for two fundamental tasks: model averaging and model selection. We found that MMI for model averaging reduced predictive uncertainty by combining multiple models (see "Results" section Bayesian MMI increases certainty in ERK activity predictions), retained the accuracy of the best individual models and was robust to perturbations in the set of models (see "Results" section MMI increases robustness to model form uncertainty). We also found that MMI predictions were more robust to additional data uncertainty than individual models nd that MMI enabled better out-of-sample forecasts (see "Results" section MMI is robust to increased training data uncertainty) Finally, when used for model selection, MMI allowed us to systematically compare many hypotheses and select the most consistent with the data (see "Results" section Subcellular ERK activity depends on Rap1 and negative feedback). While our findings are based on a set of analyses that use ERK signaling as an example problem, the implications of this work—that leveraging multiple models can improve predictions and increase robustness—are likely to have impacts across systems biology and the broader modeling community.

It is standard to choose *a good model* based on the biological assumptions and scope of the model without assessing predictive performance. In contrast to using a single model for predictions, we found that Bayesian multimodel inference has several advantages.

MMI increases the chances of making accurate predictions by weighting models based on predictive performance. Further, we found that MMI predictions show reduced uncertainty when averaged over multiple models. Therefore, the contribution to the overall uncertainty due to parameter estimation and model structure will be reduced. Posterior predictive simulations (Eq. (8) in "Methods") show the total uncertainty, which includes posterior uncertainty and data uncertainty[13]. We note that we only show posterior densities in this work to better highlight differences between models. This was especially prevalent when the models were used to predict future ERK activity (see "Results" section, MMI is robust to increased training data uncertainty). Finally, we found that MMI is robust to model mis-specifications and erroneous assumptions because it discards predictions from the *worst* models. For example, our predictions of the EGF-ERK dose-response would have been very poor if we had chosen only to use the O' 2009 model—a reasonable and well-justified choice—(Supplementary Fig. 4); however, MMI assigned nearly no weight to that model. These findings show how MMI accounts for model uncertainty by leveraging all user-specified models and can increase predictive certainty.

*Is there a best Bayesian multimodel inference method for systems biology?* Here, we found that pseudo-BMA, which is widely applicable whenever any Bayesian inference method is used[18], yielded the most accurate and certain predictions. Stacking tended to only select the model with the highest ELPD in our examples except when multiple data were included, such as in "Results" section, Subcellular ERK activity depends on Rap1 and negative feedback. It is important to note that stacking may only weight a single model when all of the models make similar predictions[50]. Finally, we found that BMA, although effective at model averaging, could substantially reduce predictive uncertainty while increasing predictive error beyond the best models (see, e.g., Supplementary Fig. 3). Additionally, BMA requires using specific samplers such as sequential Monte Carlo (SMC) or additional computations to estimate the marginal likelihood[19,22]. While pseudo-BMA tended to perform well for model averaging and is more widely applicable than BMA, both methods demonstrated benefits over utilizing single models. Furthermore, for model selection, we found that including multiple MMI methods for model selection applications increased confidence in the selected models or hypotheses (see, e.g., "Results" section Subcellular ERK activity depends on Rap1 and negative feedback). Therefore, based on our findings, we conclude that pseudo-BMA is a good general approach for Bayesian MMI and recommend using more than one method for model selection. We also note that directly comparing Bayesian and frequentist approaches can improve our understanding of the available MMI toolkit.

Notably, there are alternatives to the three MMI methods analyzed in this work. First, we chose to focus on Bayesian methods and exclude frequentist approaches that utilize metrics such as the Akaike or Bayesian information criteria[14,18] because the uncertainty quantification provided by Bayesian methods is most appropriate for the limited-data regimes in systems biology. In the Bayesian literature, methods including stochastic search variable selection[51], probability density fusion[24], and reversible jump MCMC[52] have also been proposed for MMI (see references within[19] and[18] for additional methods). Methods such as ensemble estimation[53] can yield predictions similar to MMI. Causal inference[54], and sparsity-promoting inference[55–58] can more-directly select models or mechanisms from data. A comparison between MMI-based model selection and alternative data-driven methods could further expand the tools available in systems biology.

*How much data are needed for effective multimodel inference?* While we found that MMI estimates are *robust* to the amount of training data provided, the minimum amount of required data will likely vary between applications and model sets. In this work, we used data that directly measured the QoI; however, MMI with data that do not measure the QoI directly might prove more challenging or result in

predictions with higher uncertainty. In general, we note that the certainty and accuracy of MMI predictions are directly tied to the quality of the individual model predictions, which is well-known to improve with increasing data, e.g., ref. 12. One possible approach to assessing how the amount of available data impacts MMI predictions is to include uncertainties in the model weights. For example, Yao et al.[22] utilize the Bayesian bootstrap to assess model weight uncertainty by sampling the approximate distribution of possible pseudo-BMA model weights. Furthermore, in this work, MMI is based on metrics that assess predictive performance using a single dataset, and incorporating multiple datasets can potentially improve the confidence in and accuracy of MMI predictions. Such efforts may require multi-data multimodel inference, which could take inspiration from meta-modeling to weigh datasets and models.

Future development of MMI for systems biology should focus on utilizing computationally efficient methods for parameter estimation and applying methods to learn time-varying MMI weights. MMI requires repeating Bayesian parameter estimation for every specified model, which can introduce a substantial computational burden to a modeling study (e.g., inference run times in Supplementary Table 3). Approaches to accelerate Bayesian estimation, such as variational inference[59] or Laplace approximation[28] can enable MMI with expensive-to-estimate models while providing approximate uncertainty quantification. Additionally, we presented MMI in a framework where the weights on each model do not vary over time. However, in settings with significant time-dependent changes, such as cell-fate transitions or development, one might expect that the underlying mechanism driving the biological process. Thus, the most applicable model could change over time. In those cases, extensions to the MMI framework, such as dynamic BMA[16] or sequential data assimilation with multiple models[60], can potentially learn model weights that are time-varying to allow for the MMI estimator to change over time.

As mathematical models continue to make critical contributions to studying biological systems, it is essential to account for model assumptions rigorously. While one of the gold standards in modeling is to develop a model that perfectly captures biological reality, limitations in our understanding of biological processes require us to rely on simplifying assumptions to build more tractable, simplified models. Furthermore, developing and working with complex, hyper-realistic models requires the ability to collect large quantities of very high-quality data to enable parameter estimation and make accurate predictions. Given that we often work with incomplete knowledge and limited data, we need to be able to make predictions that account for and are robust to uncertainties related to modeling assumptions. Thus, in these regimes, we propose that Bayesian multimodel inference provides a rigorous approach to handling model uncertainty that can potentially improve systems biology predictions and drive future biological discoveries whenever multiple models of the same system are available. Additionally, when we have access to large datasets, MMI could be the starting point for developing detailed, biologically realistic models by determining the components of simpler models most consistent with the data. In summary, we conclude that MMI is a powerful approach to account for modeling assumptions in systems biology and can improve predictive certainty and inform future model development.

## Methods
### Standardized EGF-inputs
To standardize the EGF input across models, we modified all model formulations to include EGF as a state variable. Then, we set the time derivative of EGF to zero to simulate a sustained EGF stimulus, which assumes that modeled reactions do not deplete extracellular EGF pools. Additionally, we define all EGF stimuli in nanomolar (nM) concentrations. To convert from concentrations in ng/mL or pg/mL to nM we assume EGF has a molecular weight of 6, 048g/mol[61]. Additionally, to convert from nM to molecules/cell, we assume a cell volume of 1

nanoliter (nL)[62]. Lastly, in models that include additional growth factors, we set the concentrations of the corresponding receptors to zero. Specifically, in Birtwistle et al. 2007[40] we fixed concentrations for H, E3, and E4, and in von Kreigsheim et al. 2009[42] we fixed the NGF and NGFR concentrations.

## Experimental data preprocessing
From live-cell fluorescence microscopy experiments, Keyes et al. report all EKAR measurements as the YFP/CFP emission ratio, $R(t)$[27]. In "Results" section, Bayesian MMI increases certainty in ERK activity predictions, we normalized each cell-wise trajectory by removing the cell-wise minimum and dividing by the difference in the cell-wise maximum and minimum. Further, in "Results" section, Subcellular ERK activity depends on Rap1 and negative feedback, we defined the normalized EKAR4 data as $\tilde{R}(t) := R(t)/\overline{R_{max}}$, where $R(t)$ is the EKAR4 emission ratio, and $\overline{R_{max}}$ is the mean (across all cells with and without Rap1 inhibition) of the maximum (in time) emission ratio. This normalization choice retains location-specific differences in ERK signaling and the effects of Rap1 inhibition. All original data are reported as measurements over individual cells; thus, to yield a single dataset with an uncertainty estimate, we computed the average and standard deviation over all cells. We found that normalization introduced substantial uncertainty in Rap1 inhibition data, so in constructing a likelihood model, we took the standard deviation of the data to be one-half of that computed directly. To ensure that the data and model predictions were on the same scale, we similarly normalized model predictions of active ERK to the trajectory-wise maximum.

## Synthetic data generation
We generated two experimentally realistic synthetic datasets to analyze MMI. First, using the H' 1996 model, we recreated the EGF-ERK dose-response curve shown in Figure 2B of ref. 30 by varying the input EGF concentration from 0.001 nM to 0.106 nM over 10 levels and computing the resulting steady-state ERK activity. The range of EGF input concentrations is also similar to experiments in ref. 63. Specifically, we normalized each response to the maximum across input levels and denoted this by the % maximal ERK activity. Further, we assumed that each measurement is subject to mean-zero independent Gaussian measurement error with a standard deviation of 0.1, similar to measurement uncertainties observed in experiments[30,63]. To account for differences in the total available ERK concentration across models and ensure that model predictions are on the same scale of the data, we computed the % maximal ERK activity by normalizing each predicted dose-response curve to the maximum ERK activity in that curve.

## Bayesian parameter estimation and sequential Monte Carlo
In this section, we provide a brief overview of Bayesian parameter estimation. For more details in the context of systems biology, see refs. 11,12, and more theory in general, see refs. 10,13.

Systems biology models use systems of *ordinary differential equations* (ODEs) to describe the rates of change in the concentration of included biochemical species. For a particular signaling pathway, we often have a set of $K$ models, $\mathfrak{M}_K = \{\mathcal{M}_1, \ldots, \mathcal{M}_K\}$, that vary in their representation of the system. Each model connects the dynamics of the state variables $\mathbf{x}_k(t) \in \mathbb{R}_+^{n_k}$, ($\mathbb{R}_+ = [0, \infty)$ are the nonnegative real numbers) which correspond to the concentration of biochemical species, to observations $\hat{\mathbf{y}}_k(t) \in \mathbb{R}_+^{m_k}$ using the system of equations

$$\frac{d\mathbf{x}_k(t)}{dt} = f_k(\mathbf{x}_k(t); \theta_k), \tag{3}$$

$$\hat{\mathbf{y}}_k(t) = h_k(\mathbf{x}_k(t); \theta_k) + \boldsymbol{\eta}(t), \quad \boldsymbol{\eta}(t) \sim \mathcal{N}(\mathbf{0}, \boldsymbol{\Gamma}), \tag{4}$$

where $f_k(\cdot; \cdot) : \mathbb{R}_+^{n_k} \times \mathbb{R}_+^{p_k} \to \mathbb{R}_+^{n_k}$, and $h_k(\cdot; \cdot) : \mathbb{R}_+^{n_k} \times \mathbb{R}_+^{p_k} \to \mathbb{R}_+^{m_k}$. The model parameters $\theta_k \in \Theta_k \subseteq \mathbb{R}_+^{p_k}$ control model predictions and

include quantities such as reaction rates and equilibrium coefficients. We additionally assume that a Gaussian measurement noise process $\boldsymbol{\eta}(t) \in \mathbb{R}^{m_k}$ with covariance matrix $\boldsymbol{\Gamma} \in \mathbb{R}^{m_k \times m_k}$ reflects uncertainty in the measurements. Beyond predictions of the observations, we assume that each model predicts a biologically relevant quantity of interest (QoI) $q(t) \in \mathbb{R}$. The QoI is predicted by a function of the internal states and parameters, $\hat{q}_k = g_k(\mathbf{x}_k(t), \theta_k)$, where $\hat{q}_k$ is the QoI prediction with model $\mathcal{M}_k$, and $g_k(\cdot, \cdot) : \mathbb{R}_+^{n_k} \times \mathbb{R}_+^{p_k} \to \mathbb{R}$.

Bayesian parameter estimation learns a probability density for the parameters of each model conditioned on the training data $p(\theta_k | d_{train}, \mathcal{M}_k)$[10,13]. The training data $d_{train} = \{\mathbf{y}^1, \ldots, \mathbf{y}^{N_{train}}\}$ consists of $N_{train}$ noisy experimental observations, and can correspond to time points $t^i$ in dynamic responses $\mathbf{y}^i = \mathbf{y}(t^i)$ or to input stimuli $u^i$ in dose-response curves $\mathbf{y}^i = \mathbf{y}(u^i)$. Bayesian estimation applies Bayes' rule

$$\underbrace{p(\boldsymbol{\theta}_k | d_{train}, \mathcal{M}_k)}_{\text{posterior}} \propto \underbrace{p(\boldsymbol{\theta}_k | \mathcal{M}_k)}_{\text{prior}} \underbrace{p(d_{train} | \boldsymbol{\theta}_k)}_{\text{likelihood}}, \tag{5}$$

which relates the *posterior probability density* of the model parameters, $p(\theta_k | d_{train}, \mathcal{M}_k)$, to the product of the *prior density* $p(\theta_k | \mathcal{M}_k)$ and the *likelihood function* $p(d_{train} | \theta_k)$. The posterior density is the probability density of the model parameters given the available training data and the model. The prior density encodes assumptions about the parameters before considering training data. The likelihood function measures the probability that model predictions match the training data and is a function of the model parameters. In most systems biology problems, we do not have a closed-form equation to evaluate the posterior density directly, so we instead must rely on methods such as *Markov chain Monte Carlo* (MCMC)[10,12,13] or SMC[64,65] to characterize the posterior through the $S$ samples drawn from it, $\{\theta_k^1, \ldots, \theta_k^S\} \sim p(\theta_k | d_{train}, \mathcal{M}_k)$.

This work uses log-normal prior densities for all unknown model parameters. To let the data inform the estimation procedure, we choose to center the mean of each prior on the logarithm of the nominal values for the unknown parameters $\theta_k^{nominal}$ and let the standard deviation be suitably large such that 95% of the probability mass of the prior is in the range $[10^{-2} \times \theta_k^{nominal}, 10^2 \times \theta_k^{nominal}]$. One can show that this corresponds to setting the prior standard deviation to 2.350. Substantial empirical evidence suggests that wide, weakly informative priors such as the lognormal prior with wide variance greatly enable sampling of the posterior density compared to completely uninformative priors such as the uniform prior[13]. Next, the likelihood function is a Gaussian density because we assume a Gaussian measurement noise in Eq. (4)[12]. We use the PyMC probabilistic programming library in Python to build statistical models efficiently and enable posterior sampling[66].

We use SMC to sample from the posterior density because it estimates the marginal likelihood without additional computation[64,65]. SMC is a particle-based sampler that sequentially adapts the particles from prior samples to posterior samples using a tempering scheme. The particles are weighted and mutated at each sampler stage with an importance sampling step. At the final stage, the weights assigned to the particles correspond to the marginal likelihood and can be averaged to provide a marginal likelihood estimate. We use the implementation provided by the PyMC Python package with the independent Metropolis-Hastings transition kernel. Additionally, we set the `correlation_threshold` parameter to 0.01 and the `threshold` parameter to 0.85. Unless otherwise noted, we run four independent SMC chains with at least 500 posterior samples per chain.

## Forward uncertainty propagation with ensemble simulation
Given $S$ samples from the posterior density, we perform ensemble simulations to propagate uncertainty forward to model predictions. Specifically, we solve the ODE model with each of the $S$ posterior samples to generate a set of ODE solutions for resulting predictive

densities. There are three important predictive densities that we use to assess predictive uncertainty. First, the *posterior push-forward density of the QoI* (also called the predictive density of the QoI) is defined as

$$p(\hat{q}_k|d_{\text{train}}, \mathcal{M}_k) := \int_{\Theta_k} g_k(\mathbf{x}_k(t), \theta_k) p(\theta_k|d_{\text{train}}, \mathcal{M}_K) \, d\theta_k \quad (6)$$

and directly propagates parametric uncertainty to the QoI. To sample from the posterior push-forward density of the QoI, we evaluate the QoI function $g_k(\cdot, \cdot)$ at each sample, yielding a set of QoI samples $\{\hat{q}_k^1, \ldots, \hat{q}_k^S\} \sim p(\hat{q}_k|d_{\text{train}}, \mathcal{M}_k)$. Next, the *posterior push-forward density of the measurements*,

$$p(\hat{\mathbf{y}}(t)_k|d_{\text{train}}, \mathcal{M}_k) := \int_{\Theta_k} h_k(\mathbf{x}_k(t)) p(\theta_k|d_{\text{train}}, \mathcal{M}_K) \, d\theta_k, \quad (7)$$

similarly propagates parametric uncertainty to the measurements. We evaluate the measurement function at each ODE solution to sample from the posterior push-forward density of the measurements. Lastly, the *posterior predictive density* is defined as

$$p(\tilde{d}_{\text{train}}|d_{\text{train}}, \mathcal{M}_k) := \int_{\Theta_k} p(d_{\text{train}}|\theta_k, \mathcal{M}_k) p(\theta_k|d_{\text{train}}, \mathcal{M}_K) \, d\theta_k, \quad (8)$$

and accounts for both parametric and data uncertainty. To obtain posterior predictive samples, we sample the posterior push-forward density of the measurements and subsequently add independent samples from the measurement noise process defined in Eq. (3). We refer the reader to ref. 31 and ref. 13 for more details on predictive densities and Bayesian model analysis in general. In this work, we use the posterior push-forward of the QoI to propagate uncertainty.

## Bayesian model averaging

BMA weighs each model with the model probability $w_k^{\text{BMA}} = p(\mathcal{M}_k|d_{\text{train}})$[19]. Notably, the model probability is the realization of a discrete probability mass function over the set of models. The model probability is computed by applying Bayes' rule a second time (the first time is for model parameter estimation) at the model level, that is

$$p(\mathcal{M}_k|d_{\text{train}}) = \frac{p(d_{\text{train}}|\mathcal{M}_k) p(\mathcal{M}_k)}{\sum_{l=1}^{K} p(d_{\text{train}}|\mathcal{M}_l) p(\mathcal{M}_l)}, \quad (9)$$

where $p(\mathcal{M}_k)$ is the prior model probability and $p(d_{\text{train}}|\mathcal{M}_k)$ is the marginal likelihood. Importantly, the marginal likelihood is included in Eq. (9) because it depends on the model, whereas we exclude the denominator in parameter estimation and replace equality with proportionality in Eq. (5). The *marginal likelihood*,

$$p(d|\mathcal{M}_k) = \int_{\Theta_k} p(d|\theta_k, \mathcal{M}_k) p(\theta_k|\mathcal{M}_k) \, d\theta_k, \quad (10)$$

quantifies the probability of observing the data under model $\mathcal{M}_k$. Direct computation of Eq. (10) can become intractable because the integral is often over a high-dimension space and thus requires approximations by Monte Carlo integration, Bridge Sampling, or SMC[19,67–69]. In this work, we use the PyMC implementation of SMC, which estimates the log marginal likelihood alongside posterior samples[64–66]. PyMC returns a log marginal likelihood estimate for each independent chain, so we take the average over all chains to generate an estimate for that model. Additionally, we assume that each model is equally probable a priori with prior model probability $p(\mathcal{M}_k) = 1/K$. We employ the log-sum exponential trick to avoid numerical overflow in evaluating Eq. (9), which involves computing the sum of the exponent of log marginal likelihoods. That is, given log marginal likelihoods, Eq.

(9) can be written abstractly as

$$p(\mathcal{M}_k|d_{\text{train}}) = \frac{\exp(x_k)}{\sum_{l=1}^{K} \exp(x_l)}, \quad (11)$$

where $x_k = \log\big(p(d_{\text{train}}|\mathcal{M}_k) p(\mathcal{M}_k)\big)$ can be quite large and thus Eq. (11) is susceptible to overflow errors. We can rearrange Eq. (11) to

$$p(\mathcal{M}_i|d_{\text{train}}) = \exp\left[x_i - \log \sum_{l=1}^{K} \exp(x_l)\right], \quad (12)$$

where we use the log-sum-exponential trick

$$\log \sum_{l=1}^{K} \exp(x_l) = c + \log \sum_{l=1}^{K} \exp(x_l - c), \quad (13)$$

with $c = \max[x_1, \ldots, x_K]$ for numerical stability.

## Pseudo-Bayesian model averaging

Pseudo-Bayesian model averaging weighs models based on the expected predictive performance of future data measured with the ELPD. Similar to Akaike-type weighting, which uses the AIC instead of the ELPD[14], we define the scalar pseudo-BMA weights as

$$w_k^{\text{pBMA}} := \frac{\exp(\widehat{\text{ELPD}}_{\mathcal{M}_k})}{\sum_{i=1}^{K} \exp(\widehat{\text{ELPD}}_{\mathcal{M}_k})}, \quad (14)$$

given an estimate of the ELPD for each model $\widehat{\text{ELPD}}_{\mathcal{M}_k}$. Further, we stabilize Eq. (14) with the log-sum-exponential trick as in the previous section. To account for uncertainty in the ELPD estimates, we use the Bayesian bootstrap as was done in refs. 22,70.

We adapt the definitions from ref. 18 to introduce approximations to the ELPD that can be used for pseudo-BMA. First, we let the training data $d_{\text{train}} = \{\mathbf{y}^1, \ldots, \mathbf{y}^{N_{\text{train}}}\}$ consist of statistically independent data points. It follows that $p(d_{\text{train}}|\theta_k, \mathcal{M}_k) = \prod_{i=1}^{N_{\text{train}}} p(\mathbf{y}^i|\theta_k, \mathcal{M}_k)$ due to the independence of the data. The ELPD of model $\mathcal{M}_k$ is defined as

$$\text{ELPD}_{\mathcal{M}_k} := \sum_{i=1}^{N_{\text{train}}} \int \log p(\tilde{\mathbf{y}}_i|d_{\text{train}}, \mathcal{M}_k) p_{\text{true}}(\tilde{\mathbf{y}}_i) \, d\tilde{\mathbf{y}}_i, \quad (15)$$

and quantifies the expected predictive performance of the model compared to the true data-generating distribution $p_{\text{true}}(\tilde{\mathbf{y}}_i)$. The *log posterior predictive density* for model $\mathcal{M}_k$ and new data point $\tilde{\mathbf{y}}_i$ is defined as

$$\log p(\tilde{\mathbf{y}}_i|d_{\text{train}}, \mathcal{M}_k) := \log \int_{\Theta_k} p(\tilde{\mathbf{y}}_i|\theta_k, \mathcal{M}_k) p(\theta_k|d_{\text{train}}, \mathcal{M}_k) \, d\theta_k \quad (16)$$

and measures the log probability of observing the new data point. Given $S$ posterior samples $\hat{\theta}_k^s$, the log posterior predictive density can be approximated with

$$\log p(\tilde{\mathbf{y}}_i|d_{\text{train}}, \mathcal{M}_k) \approx \log \frac{1}{S} \sum_{i=1}^{S} p(\tilde{\mathbf{y}}_i|\hat{\theta}_k^s, \mathcal{M}_k). \quad (17)$$

In general, we do not know the true probability of the data $p_{\text{true}}(\tilde{\mathbf{y}}_i)$, so we must rely on approximations to the ELPD such as the leave-one-out cross-validation estimator (LOO-CV)[18], which we use in this work, or the widely applicable information criterion (WAIC)[71].

The simplest approximation to the ELPD, called the *sample-approximated log pointwise predictive density* (LPD), is simply the sum

of Eq. (17) over all data points,

$$\widehat{\mathrm{LPD}}_{\mathcal{M}_k} = \sum_{i=1}^{N_{\mathrm{train}}} \log \frac{1}{S} \sum_{s=1}^{S} \mathrm{p}(\mathbf{y}_i|\hat{\theta}_k^s, \mathcal{M}_k); \qquad (18)$$

however, the LPD is known to overestimate the ELPD[18]. Thus, we can use LOO-CV to approximate the ELPD as

$$\widehat{\mathrm{ELPD}}_k^{\mathrm{LOO}} := \sum_{i=1}^{n} \log \mathrm{p}(\mathbf{y}_i|d_{\mathrm{train}}^{-i}, \mathcal{M}_k), \qquad (19)$$

where the *leave-one-out predictive density* is defined as

$$\mathrm{p}(\mathbf{y}^i|d_{\mathrm{train}}^{-i}, \mathcal{M}_k) := \int \mathrm{p}(\mathbf{y}^i|\theta_k, \mathcal{M}_k)\mathrm{p}(\theta_k|d_{\mathrm{train}}^{-i})\,\mathrm{d}\,\theta_k. \qquad (20)$$

Here, $d_{\mathrm{train}}^{-i} := \{\mathbf{y}^1, \ldots, \mathbf{y}^{i-1}, \mathbf{y}^{i+1}, \ldots, \mathbf{y}^{N_{\mathrm{train}}}\}$ indicates all of the training data $d_{\mathrm{train}}$ excluding point $\mathbf{y}^i$. We refer the reader to Vehtari et al.[18] for further details and motivation. This work uses the LOO-CV-based approximation over the WAIC-based approximation to the ELPD because both approximations provided similar weights for each model (for example, see Supplementary Fig. 12). However, direct computation of Eq. (20) can be prohibitively computationally expensive because it requires repeating parameter estimation for each held-out data point. To enable efficient computation of the LOO-CV ELPD estimator, we use Pareto smoothed importance sampling[18,72], the details of which we omit for brevity. We denote the PSIS-LOO-CV estimator of the ELPD as PSIS-LOO-CV.

In this work, we use the ArviZ Python library[73] to compute the PSIS-LOO-CV and the resulting model weights. Given a dictionary of inference data objects with log-likelihood samples for each model, we use the `Arviz.compare()` function with the `ic` parameter set to `loo` and the `method` parameter set to `BB-pseudo-BMA`. We use default settings for all computations. To ensure that PSIS-LOO-CV accurately estimates the LOO-CV estimator, we compared PSIS-LOO-CV to direct leave-one-out cross-validation (LOO-CV) for the S' 2014 model using synthetic dose-response data (Supplementary Fig. 4a). Notably, the ELPD predictions for PSIS-LOO-CV and LOO-CV are within five percent, predicting −5.69 and −5.43, respectively.

## Stacking of predictive densities

Stacking of predictive densities[22] aims to assign scalar weights $w_k^{\mathrm{stack}}$ to each model such that the new estimator is optimal according to a specified optimality criterion. The stacking optimization problem

$$\max_{\{w_k^{\mathrm{stack}}\}} S\left(\sum_{k=1}^{K} w_k^{\mathrm{stack}} \mathrm{p}(\tilde{d}|d_{\mathrm{train}}, \mathcal{M}_k), \mathrm{p}_{\mathrm{true}}(\tilde{d})\right) \qquad (21)$$

aims to maximize a score, $S(\cdot, \cdot) : \mathcal{P} \times \Omega \to \mathbb{R}$, where $\mathcal{P}$ is a probabilistic forecast and $\Omega$ is the sample space on which the true predictive density is defined (see ref. 22 for a more detailed definition). The score is computed between the consensus predictive density evaluated at new data $\tilde{d}$ and the true predictive density. However, the true predictive density is unknown, so Yao et al.[22] suggest using a logarithmic scoring rule and replacing the predictive density evaluated at new data with the LOO estimator. The approximate stacking optimization problem becomes

$$\max_{\{w_k^{\mathrm{stack}}\}} \frac{1}{n} \sum_{i=1}^{n} \log \sum_{k=1}^{K} w_k^{\mathrm{stack}} \mathrm{p}(\mathbf{y}_i|d_{\mathrm{train}}^{-i}, \mathcal{M}_k), \qquad (22)$$

where the LOO-CV predictive density $\mathrm{p}(\mathbf{y}_i|d_{\mathrm{train}}^{-i}, \mathcal{M}_k)$ can again be estimated with PSIS-LOO-CV as in the previous section.

We again use the ArviZ Python package[73] to compute the stacking weights in this work. Specifically, we use the `Arviz.compare()`

function with the `ic` parameter set to `loo` and the `method` parameter set to `stacking`. All other parameters are set to the defaults.

## Numerical solution of ODEs and steady-state simulation

As systems biology models are often stiff ODE systems, we solve all systems of ordinary differential equations with the Kvaerno 4/5 implicit Runge-Kutta method that is implemented in the Diffrax Python package and is well-suited for integrating stiff systems of ODEs[74,75]. Unless otherwise stated, we set the maximum number of solver steps, `max_steps`, to $6 \times 10^6$. To account for potential stiffness, in conjunction with the implicit scheme, we use a PID controller-based method for adaptive time stepping[76–78] with tolerances `atol=1e-6` and `rtol=1e-6` for all problems unless otherwise noted in Supplementary Table 2. Such tolerances are suggested to provide sufficient accuracy when solving most systems. We run the solver from the initial condition to the desired time point to obtain the entire solution over time. However, we choose the following methods to balance accuracy and computational efficiency to obtain each model's steady-state solution. The first approach runs the ODE solver until the solution and time derivative satisfy $\|\mathrm{d}\mathbf{x}(t)/\mathrm{d}t\|_2 \le \mathrm{atol} + \mathrm{rtol}\|\mathbf{x}(t)\|_2$, where $\|\cdot\|_2$ is the standard vector two-norm. To evaluate this criterion, we use a Diffrax `SteadyStateEvent` with tolerances `atol=1e-5` and `rtol=1e-6` for all problems unless stated otherwise in Supplementary Table 2. The second approach uses Newton's method to directly solve $\mathrm{d}\mathbf{x}(t)/\mathrm{d}t = 0$. We use the standard Newton method implemented in the Optimistix Python library[79] with `max_steps = 100` and tolerances `atol=1e-10` and `rtol=1e-10`. To mitigate convergence issues, we initialize the Newton solver at a point along the solution of the ODE by running the previous ODE-based steady-state method with crude tolerances `atol=1e-5` and `rtol=1e-5` for all problems unless stated otherwise in Supplementary Table 2.

## Structural identifiability analysis

We perform an a priori local structural identifiability analysis to reduce the unknown model parameters with unique values given the observables. We refer the reader to refs. 80,81 for background and mathematical details on structural identifiability analysis. We perform a local analysis instead of a global analysis as suggested in our previous work[12], because computing global identifiability proved computationally intractable for larger models such as H' 2005 and B' 2007. Specifically, we used the structural identifiability analysis method from ref. 81 that is implemented in the StructuralIdentifiability.jl package in the Julia programming language. We use default settings for the `assess_local_identifiability()` function, and set the probability of correctness to $p = 0.99$ for all models except Birtwistle et al. 2007[40], for which we take $p = 0.95$ to improve computational efficiency. Additionally, due to further software limitations, we exclude any parameters in exponents from the identifiability analysis. If these parameters are integers, we fix them to their nominal value, otherwise, we set them to 1.0. Specifically, for Hornberg et al. 2005[39] we fix $n$ to 1.0, and for von Kreigsheim et al. 2009[42] we fix $k_{57}$, $k_{61}$, $k_{64}$, $k_{66}$, $k_{70}$, and $k_{72}$ to 1.0.

## Global sensitivity analysis

We use the Morris screening method for global sensitivity analysis[82,83]. First, we assume that all parameters vary independently and can take on values in the range $[10^{-2} \times \theta^{\mathrm{nominal}}, 10^2 \times \theta^{\mathrm{nominal}}]$ (nominal values are listed in Supplementary Materials). We then use the steady-state activated ERK concentration as the quantity of interest for models that predict sustained ERK activation at the nominal parameter values, i.e., H' 1996, O' 2009, S' 2014, R' 2015, and K' 2017, or the maximal activated ERK concentration as the QoI for models that predict a transient activation, i.e., K' 2000, L' 2000, H' 2005, B' 2007, and vK' 2009. Additionally, we assume that the EGF concentration is fixed at 0.1 nM

for all models. To perform the Morris screening, we use the SALib Python package[84,85] with the method of Morris sampler (`SALib.sample.morris.sample()`) and analysis (`SALib.sample.morris.analyze()`) functions with default settings. We draw 256 samples per parameter direction for all models except H' 2005 and B' 2007, for which we use 30 and 10 samples, respectively. Morris screening provides two measures of sensitivity, the mean of the distribution of the absolute value of elementary effects $\mu^*$ and the standard deviation of the distribution of elementary effects $\sigma$[83]. We refer the reader to refs. [46,83] for more details. We assume that parameters are influential to ERK activation when $\mu_i^*/\max\{\mu_i^*\} > 0.1$ or $\sigma_i/\max\{\sigma_i^*\} > 0.1$.

**Error and uncertainty metrics**

We measure predictive error with either the root mean square error (RMSE) or the relative error. Given a prediction $\hat{\mathbf{y}}$ and a reference (ground truth) $\mathbf{y}$, the *RMSE* is defined as,

$$\text{RMSE} := \frac{\sqrt{\sum_{i=1}^N (\hat{y}_i - y_i)^2}}{N}, \tag{23}$$

where $\mathbf{y} \in \mathbb{R}^N$, and $y_i$ is the $i$th element of $\mathbf{y}$. We additionally define the *relative error* as

$$\text{Relative Error} := \frac{||\hat{\mathbf{y}} - \mathbf{y}||_2}{||\mathbf{y}||_2}, \tag{24}$$

where the norm is the standard vector two-norm, $||\mathbf{a}||_2 = \sqrt{\sum_{i=1}^N a_i^2}$.

We measure predictive uncertainty using the average width of the 95% credible interval. Given a set of predictive samples prediction, $\hat{\mathbf{y}}^1, \ldots, \hat{\mathbf{y}}^S$, we define the *95% credible interval* as the interval between the 2.5th and 97.5th percentiles,

$$95\% \text{ credible interval} := [P_{2.5}, P_{97.5}], \tag{25}$$

where $P_i$ denotes the element-wise $i$th percentile of the set of samples. Each element of $P_i$ is the percentile of the corresponding set of elements from the set of vector-valued predictive samples. We define the average width of the 95% credible interval as the mean over all elements of the difference between $P_{97.5}$ and $P_{2.5}$.

**Statistical comparison of probability densities**

We compared probability densities using the Kolmogorov–Smirnov test (K–S Test) or the Mann–Whitney $U$-test with two-sided hypotheses. All statistical comparisons were performed using the Scipy Python library.

**Reporting summary**

Further information on research design is available in the Nature Portfolio Reporting Summary linked to this article.

## Data availability

All data supporting the findings of this study are available in the paper and the supplement. Original subcellular ERK activity data are available from Keyes et al. [27] and the associated supplementary materials. Source data are provided with this paper.

## Code availability

All code required to reproduce this work is available on Zenodo at https://doi.org/10.5281/zenodo.15129141 (ref. [86]).

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

## Acknowledgements

N.L.S. acknowledges support from the National Institute of Biomedical Imaging and Bioengineering (NIBIB) of the National Institutes of Health (NIH; https://www.nibib.nih.gov) under award number T32EB9380 and a UCSD Sloan Scholar Fellowship from the Alfred P. Sloan Foundation (https://sloan.org). P.R. acknowledges support from Air Force Office of Scientific Research (AFOSR; https://www.afrl.af.mil/AFOSR/) Multi-disciplinary University Research Initiative (MURI) grant FA9550-18-1-0051. N.L.S. and P.R. acknowledge support Wu Tsai Human Performance Alliance at UCSD.

## Author contributions

N.L.S. led the study design, conducted all analyses, prepared all of the figures and wrote the initial draft of the manuscript. P.R. and B.K. supervised the project and contributed to study design and manuscript preparation. J.Z. supervised original collection of experimental data used in this work. All authors reviewed the manuscript

## Competing interests

P.R. is a consultant for Simula Research Laboratories in Oslo, Norway, and receives income. The terms of this arrangement have been reviewed and approved by the University of California, San Diego in accordance with its conflict of interest policies. All other authors declare no competing interests.
