## [Transparent Peer Review file · Nature Communications]

Increasing certainty in systems biology models using Bayesian multimodel inference

Corresponding Author: Professor Padmini Rangamani

Version 0:

Reviewer comments:

Reviewer #1

(Remarks to the Author)

The paper presents Bayesian multimodel inference (MMI) framework for systems biology. This scheme, applied in the ERK signaling field, uncovers subcellular location-specific variations in ERK activity.

It compares three weight selection methods—BMA, pseudo-BMA, and stacking—allowing MMI to be applied to biological data; the experimental design looks robust; MMI's applicability for intracellular signaling provides insights into the mechanisms of the Rap1 pathway and ERK negative feedback. However, there are several issues that should be addressed.

1. While the application of MMI framework to intracellular signaling is noteworthy, the methods themselves—BMA, pseudo-BMA, and stacking—are not novel, i.e. the theoretical contributions are weak.
2. It reveals the combined mechanism of Rap1 signaling and ERK feedback but doesn't compare MMI to the methods like ensemble learning or causal inference, which could also capture similar/other interactions. A comparison would strengthen the case for MMI's superiority.
3. On page 4, the reference to the QoI for ERK signaling is unclear. The paper should explain the biological significance of the EGF-ERK dose-response curve and the time-dependent trajectory of EGF-induced ERK activity.
4. In the introduction to pBMA in the Results section, a brief mention of the ELPD formula, explaining that higher values indicate better model performance, would help readers interpret the model comparison in the subsequent figures.

Reviewer #2

(Remarks to the Author)

The manuscript "Increasing certainty in systems biology models using Bayesian multimodel inference" by Linden-Santangeli et al. presents a framework for performing inference using "multimodel inference" (MMI). The authors present a brief review of MMI, and through a case study of ERK signaling, demonstrate the advantages of MMI over alternatives.

The manuscript is very well written, and likely a useful contribution. My biggest criticism, however, is that it is unclear from the outset how the framework that the authors develop is different from existing MMI frameworks. Specifically, it is difficult to tell whether the paper should be considered a review of MMI, which is already well developed, that focusses on one case study, or whether the authors have significantly expanded upon existing MMI frameworks and are presenting a new statistical method.

I also have a few other comments and concerns, that I believe should be addressed before the paper could be considered for publication.

1. Choice of model for a case study. I agree that ERK is potentially a good case study, as there are often many potential candidate models that can describe any amount of data equally well. However, a drawback (of either the choice of system, or the implementation) is that all the models can be considered to be nested inside a larger model. Thus, the pure Bayesian approach to MMI would be equivalent to performing inference on this larger model, subject to a spike-slab type prior which restricted model parameters such that only the chosen k models can arise. The authors don't appear to acknowledge this, and I can't see why, if "prediction uncertainty" is the main goal, a MMI-based approach would be better than simply

performing inference on the "super-model". I push back on the authors suggestion that MMI is critical to select the best mechanism for the data as it avoids making a priori assumptions:- by its very nature, MMI effectively does make very strong a priori assumptions (albeit, perhaps less so than many studies that utilise a single model).

2. On the above comment, it is perhaps, then, no surprised that the MMI approach performs well for the case study considered, as all models can fit the data well. It would be a useful (perhaps even critical) comparison if the authors included several models that were not able to capture the data.

3. In the hypothetical limit than an infinity of models (or a large amount) are used, is it possible that the MMI approach would behave similarly to a Gaussian-process-like (or purely machine learning-like) method?

4. The authors focus throughout on "reducing uncertainty". I am not convinced that this is the best metric to assess the suitability of a statistical method. A more convincing goal would be to establish the approach that most accurately quantifies uncertainty. For example, the "best framework", by the authors current methodology, could be one that predicted very little variance in model predictions. However, this would never be expected from a finite amount of experimental data. Thus (following from comment (1), above), I am not yet convinced (but could be) from the authors present work that alternatives to Bayesian-MMA, which has a very clear statistical justification, perform better. To test this, I would imagine that one would need to predict outside the data range (i.e., different inputs) and validate on unseen experimental (or synthetic) data.

5. (minor) The authors should be more careful in their use of statistically loaded words like "accurate and certain predictions" (this could, potentially, have confused my reading the the paper and led to comment (4) above).

Reviewer #3

(Remarks to the Author)

In this paper, the authors employ Bayesian multi-model inference (MMI) to establish and test a framework to reduce uncertainties and improve the predictive capabilities of a suite of models employed in systems biology. Specifically, they employ ten models of extracellular-regulated kinase (ERK) signaling to pose and illustrate the framework. However, the framework is very generally and will equally apply to a range of deterministic models employed to predict other phenomena in systems biology. They focused on two primary questions: (i) what techniques can be employed to reduce uncertainty on model predictions, and (ii) what are techniques to reduce model choice uncertainty to improve the certainty of model predictions.

To address these issues, they cite previous work in which they employed sensitivity analysis to determine model parameters which significantly influence considered model responses. They subsequently applied Bayesian inference to calibrate parameters in the ten models using training data. The primary novelty is the main step in which they employed Bayesian multi-model inference to combine specified models to reduce uncertainty and improve the resulting model predictions. For this final step, they employed three methods for choosing the weights: Bayesian model averaging (BMA), pseudo-Bayesian model averaging (pseudo-BMA), and stacking of predictive densities (stacking). Using this framework, they demonstrated the capability of significantly decreasing uncertainty and improving model predictions in the context of the ten ERK models. In the process, they address the following broad questions:

- Is there a best Bayesian multi-model inference methodology for systems biology?
- How much data are needed for effective multi-model inference?

Overall, this is a very strong paper and use of Bayesian multi-model inference constitutes an important framework for improving the accuracy of models employed for systems biology. Whereas demonstrated in this context, the use of Bayesian multi-model inference is equally important in other biological, physical, and engineering applications employing deterministic predictive models; e.g., ODE or PDE models. Moreover, the authors very carefully detail and highlight both the implementation of the framework and its ramification for the ERK models. These constitute significant strengths of the paper and augment its impact in the field.

There is one substantive comment and several more minor comments which should be addressed before the paper is accepted for publication.

Substantive Comment

This comment pertains more to the presentation style than the content. The authors are correct that the primary contribution of the paper is the use of Bayesian multi-model inference (MMI) in the context of systems biology rather than development a fundamental statistical framework for Bayesian MMI. To establish this contribution, they very carefully detail the advantages of the framework with the three techniques for choosing modeling weights in the context of the ten ERK models. In this sense, the paper is very strong. The potential issue is that this attention to detail for the specific models may reduce the impact of the paper for readers in other areas of systems biology or science. For example, the discussion on page 12 very clearly establishes the impact of the MMI techniques for the ERK models but it may be difficult to follow for readers from other research areas. My recommendation to address this would be to slightly refine some of the discussion to further highlight high-level contributions of the framework in the context of general systems biology models, as indicated in the title. This might be achieved by rewording some of the discussion or further highlighting key concepts.

Minor Comments

- The authors cite previous work in which they used sensitivity analysis to determine influential parameters for the analysis in this paper. It would be good to include a sentence indicating the number of original model parameters and the reduced

number employed in this analysis. This would highlight the importance of this step to readers.

- The authors detail on page 15 the manner in which they generate unbiased Gaussian observation errors. It would be good to indicate if the errors are additionally independent.
- On page 19, it was stated that they used a 4/5 implicit Runge-Kutta method to numerically integrate the ODE systems. It would be advantageous to very briefly discuss whether these systems are stiff and, if so, why they did not employ a stiff ODE routine to improve efficiency.
- There seemed to be some confusion regarding the nature of reported uncertainty intervals for responses. In the Discussion section on pages 16-17, they noted the use of posterior push-forward intervals for the Quantity of Interest (QoI) when sampling parameter uncertainty and posterior predictive distributions when sampling both parameter and data uncertainty. However, in earlier discussion and figure captions, they appear to refer to the former as credible intervals, which is more common when referring to intervals in the QoI due to parameter uncertainties. The consistency of these terms should be checked.
- The notation employed for citing other works was confusing and somewhat nonstandard; e.g., H' 1996 for Huang and Ferrell 1996. Unless this is required by the journal, it would probably aid readers to include the more complete references.
- It is common to additionally perform Bayesian model averaging using stochastic search variable selection (SVSS). This does not need to be discussed but it would aid readers to include a citation; e.g., B.J. Reich and S.K. Ghosh, Bayesian Statistical Methods, CDC Press, 2019.

Note that these comments are all minor and can be easily addressed with minimal additional discussion.

In summary, the use of Bayesian multi-model inference to develop a framework to improve the accuracy and predictive capabilities of systems biology models is timely and important. Moreover, the authors demonstrate that the considered framework isolates and reduces uncertainties pertaining to the suite of considered models. Finally, they demonstrate that the resulting model predictions are robust with regard to uncertainty in experimental and training data. These constitute significant strengths of the paper. However, the noted issues should be addressed before the paper is considered for publication in Nature Communications.

Reviewer #4

(Remarks to the Author)

The authors have conducted a complex study of multi-model inference on biological cellular signaling data, comparing multiple existing weighting methods. The paper highlights the challenges of combining models to fit to biological signaling data. Overall, they show that multi-model inference increases dynamic predictions and decreases uncertainty on those predictions, primarily on in-sample data. They then use their framework to evaluate a small subset of ERK signaling pathway models showing that models built to be consistent with existing hypothesis can fit the data. Overall, I think the paper is a sound contribution to the field of mathematical modeling and inference in biological systems. I believe the overall approach of multi-model inference is a promising and necessary direction. However, I had some issues with the strength of the claims made by the authors and believe either some re-wording or additional analysis is needed to strengthen the paper and ensure the evidence supports the claims.

Major Revisions:

More clarity on what is novel. Specifically, the authors use language indicating that the framework is novel, but it's unclear which components are new. Is this a methods comparison paper? A new methods paper? Are the methods for propagating uncertainty new? A biology paper? It seems that it is mostly a methods comparison/demonstration paper, showing how multiple existing methods can be combined to analyze complex biological problems. It is not clear what biological insight is new (see last revision suggestion).

What is the connection/expectations of predictions when training on one data-set vs another? Are different parameters estimated? How different are the models across data sets? It seems that very little out of sample prediction was done (only some projection in Figure 4). Is it possible to train on one type of data and predict on another? Or are the underlying fit models/mechanisms expected to be different between data-sets? If so, what is the potential for predictive modeling using MMI? Can it only be used to infer parameters and try to draw conclusions on each individual dataset?

I would like to see more insight/analysis of structural/mechanistic details in models. The authors start out with the question : "(1) How can we quantify the effects of uncertainty in the model formulation, called model uncertainty, on model predictions?" but do not provide enough detail of the variation in model formulation or it's effects on predictions to answer this question. The lack of visualization makes much of the text around the different models and signaling pathway structure difficult to follow. It would be helpful to have some schematic of the ERK signaling pathway which visually displays differences in different models in terms of their mechanistic complexity (at some course-grained level). Why were specific model selected or weighted more heavily? Is that because they are more complex? Or contain necessary mechanisms? The authors note they selected the models based on the idea that they contain different mechanisms, but do not follow up on why specific models may be weighted more heavily for one type of data-set or another. Generally, the presentation across models are difficult to follow because they are either high level (showing uncertainty on plots), complex (list of parameter values in the supplement), or difficult to connect with a diagram of mechanism (description in the text).

In a few places in the text the authors discuss model selection vs model weighting. In some cases, model selection is presented as undesirable (pg 4 Line 116), but in others (such as the discussion of localization) model selection is the goal.

Can the authors clarify when one would want to select a specific model vs train a weighted ensemble? And what the goals for each section of their paper is? In particular, please reconcile the idea that stacking produces undesirable results with the over all goal of selecting the most likely mechanisms.

The logic of the localized ERK activity dependence seems circular. Only parameters associated with Rap1 and negative feedback were allowed to vary between locations. These parameters were chosen because of previous findings in the literature. The authors showed that models with only these parameters varying could be fit to the data, but they do not test whether any other mechanisms could account for the differences between cytoplasmic and plasma membrane ERK activity. To do this, the authors would need to show that no other set of parameters could reproduce the behavior. While this shows that the MMI can find a consistent fit to existing hypothesis it does not show "... MMI would select the models that best reflect the hypotheses that agree with the available data." (pg 12 line 347-348), "Location-specific Rap1 signaling and ERK negative feedback were necessary to predict ERK activity's sub-cellular variability." (pg 12 Line 349- 350), or "Thus, these results show that ERK negative feedback and Rap1 activity are necessary for location-specific ERK activity... These findings suggest that cytoplasmic ERK activity depends on ERK negative feedback, while plasma membrane ERK activity depends on Rap1. Based on the MMI-selected hypothesis, we propose a new ERK negative feedback- and Rap1- dependent model for sub-cellular variability in ERK activity (Figure 5F). Here, MMI was critical to selecting the best mechanism from the available data without making any a priori assumptions that a single hypothesis would be best." (pg 12 line 385 – 391).

My main complaint here is that a small subset of mechanisms were pre-selected based on existing literature. It is not clear what the authors mean that "MMI-selected" hypothesis. The MMI selects between models but it does not select between the feedback, Rap1 activity and all other changes that could be made. Interestingly the authors note that the initial models (Fig 2B), allowed all parameters to vary between cytoplasm and plasma membrane. However, they merely claim that allowing all parameters to vary between compartments is not biologically feasible rather than analyzing whether the variance in the parameters when all are allowed to be free suggests either the same mechanisms identified in the literature or alternatives. It would be interesting to know if the less restricted models suggest something new. More generally, the authors need to clarify what was learned specifically from MMI vs from pre-selection based on previous literature and be precise in their conclusions.

Minor comments:

Unclear what is meant by "comparable model representations" line 145 pg 4
Figure S5D why does excluding the worst fit model seem to increase the RMSE?

Version 1:

Reviewer comments:

Reviewer #2

(Remarks to the Author)

I thank the authors for addressing my original concerns. I have no further comments to add.

(Remarks on code availability)

The code is well commented and easy to download. I have not reviewed or ran the code as I do not use Python.

Reviewer #3

(Remarks to the Author)

In this paper, the authors employ Bayesian multi-model inference (MMI) to establish and test a framework to reduce uncertainties and improve the predictive capabilities of a suite of models employed in systems biology. Specifically, they employ ten models of extracellular-regulated kinase (ERK) signaling to pose and illustrate the framework. However, the framework is very generally and will equally apply to a range of deterministic models employed to predict other phenomena in systems biology. They focused on two primary questions: (i) what techniques can be employed to reduce uncertainty on model predictions, and (ii) what are techniques to reduce model choice uncertainty to improve the certainty of model predictions.

To address these issues, they cite previous work in which they employed sensitivity analysis to determine model parameters which significantly influence specified model responses. They subsequently applied Bayesian inference to calibrate parameters in the ten models using training data. The primary novelty was the main step in which they employed Bayesian multi-model inference to combine specified models to reduce uncertainty and improve the resulting model predictions. For this final step, they employed three methods to choose the weights: Bayesian model averaging (BMA), pseudo-Bayesian model averaging (pseudo-BMA), and stacking of predictive densities (stacking). Using this framework, they demonstrated the capability of significantly decreasing uncertainty and improving model predictions in the context of the ten ERK models. In the process, they address the following broad questions:

- Is there a best Bayesian multi-model inference methodology for systems biology?
- How much data are needed for effective multi-model inference?

Overall, this is a very strong paper and use of Bayesian multi-model inference constitutes an important framework for improving the accuracy of models employed for systems biology. Whereas demonstrated for models in systems biology, the use of Bayesian multi-model inference is equally important in other biological, physical, and engineering applications employing deterministic predictive models. Moreover, the authors very carefully detail and highlight both the implementation of the framework and its ramification for the ERK models. These constitute significant strengths of the paper and augment its impact in the field. Finally, the authors have adequately addressed the issues which I detailed for the original submission.

In summary, the use of Bayesian multi-model inference to construct a framework to improve the accuracy and predictive capabilities of systems biology models is timely and important. Moreover, the authors demonstrate that the considered framework isolates and reduces uncertainties pertaining to the suite of considered models. Finally, they demonstrate that the resulting model predictions are robust with regard to uncertainty in experimental and training data. I now recommend that the paper be accepted for publication in Nature Communications.

(Remarks on code availability)

Reviewer #4

(Remarks to the Author)

Thank you to the authors for their extensive consideration of my comments. I am impressed with the exercise they undertook to explore additional models and clarify whether multiple mechanisms can fit the data.

I encourage the authors to include the out-of-sample prediction figure in the supplement. I do not think it is necessary, but I find it satisfying that some models do very well. They appear to have low confidence, but the predictions of the dose response curve are within the error bars for the "better models," which I think is quite impressive. I would have been troubled if they were all wrong or if the confidence intervals were overly tight for this type of out-of-sample test. That said, if the more realistic usage is projected forward in time, I understand why they may not want to include unrealistic usage.

I support the publication of the manuscript.

(Remarks on code availability)

DEPARTMENT OF PHARMACOLOGY
UCSD SCHOOL OF MEDICINE

9500 GILMAN DRIVE
LA JOLLA, CALIFORNIA 92093-0404
FAX: (858) 534-7029

PADMINI RANGAMANI
PROFESSOR, PHARMACOLOGY
PROFESSOR, MECHANICAL AND AEROSPACE ENGINEERING
UNIVERSITY OF CALIFORNIA, SAN DIEGO
CELLULAR AND MOLECULAR MEDICINE EAST, ROOM 2050
prangamani@health.ucsd.edu (858) 534-4046

April 9, 2025

Ref: Manuscript NCOMMS-24-69628 (“Increasing certainty in systems biology models using Bayesian multimodel inference”)

Dear Reviewers,

We would like to thank the reviewers for providing insightful feedback and suggestions. These comments helped us to substantially improve our manuscript. In what follows, we provide a summary of the changes we made and a point-by-point response to the individual comments made by the reviewers. Our responses and subsequent changes to the manuscript are provided below each reviewer comment.

In this document and the revised version of the manuscript, we colored changes according to each reviewer as follows: **Reviewer 1 is in green**, **Reviewer 2 is blue**, **Reviewer 3 is cyan**, and **Reviewer 4 is magenta**. The original reviewer comments are replicated here in black.

Point-by-point response:

Novelty of this work—addresses comments by all reviewers:

We thank all four reviewers for their comments and suggestions regarding the novelty of this work. We agree with reviewer # 3, in that “the primary contribution of the paper is the use of Bayesian multi-model inference (MMI) in the context of systems biology”. However, we understand how our original presentation of MMI in the context of systems biology as a *novel framework* led to possible confusion about the novelty of this work. We have made the following changes to be more specific about the novel aspects and mitigate confusion regarding our presentation.

- We have extensively revised the paper to present the work as a *novel application and analysis of MMI in the context of system biology* and to remove usage of wording such as *novel framework*. (Addresses: Reviewer comment 1, main criticism from Reviewer 2, substantive comment from Reviewer 3, and Reviewer 4 major comment 1) For example, see the updated caption for Figure 1.
- We have revised the discussion to provide more context regarding the benefit of MMI in a broader modeling context. For example, see lines 417–420.
- We have refocused the discussion on the benefits of MMI to emphasize how it enables the handling of model form uncertainty rather than necessarily focusing on uncertainty reduction.

Reviewer #1:

The paper presents Bayesian multimodel inference (MMI) framework for systems biology. This scheme, applied in the ERK signaling field, uncovers subcellular location-specific variations in ERK activity. It compares three weight selection methods—BMA, pseudo-BMA, and stacking—allowing MMI to be applied to biological data; the experimental design looks robust; MMI's applicability for intracellular signaling provides insights into the mechanisms of the Rap1 pathway and ERK negative feedback.

We thank the reviewer for carefully assessing our work and acknowledging the strengths therein.

However, there are several issues that should be addressed.

1. While the application of MMI framework to intracellular signaling is noteworthy, the methods themselves—BMA, pseudo-BMA, and stacking—are not novel, i.e. the theoretical contributions are weak.

Please see our comments to the editor above addressing the novelty of this work.

2. It reveals the combined mechanism of Rap1 signaling and ERK feedback but doesn't compare MMI to the methods like ensemble learning or causal inference, which could also capture similar/other interactions. A comparison would strengthen the case for MMI's superiority.

Indeed, ensemble learning and causal inference could also be applied in a similar setting. However, we aimed to keep this work focused on introducing MMI and three specific algorithms. We use Figure 6 (previously Figure 5; shows the analysis of Rap1 and ERK feedback) as an example of how MMI is useful rather than investigating and comparing all applicable methodologies. We added the possibility of using ensemble learning and causal inference in the Discussion section on lines 459–460 of the revised manuscript.

3. On page 4, the reference to the QoI for ERK signaling is unclear. The paper should explain the biological significance of the EGF-ERK dose-response curve and the time-dependent trajectory of EGF-induced ERK activity.

Thank you for the helpful suggestion. We now define these QoIs in the first results section of the manuscript (lines 91–98). We have also expanded the discussion on the biological significance of dose-response curves and time-dependent trajectories throughout the manuscript.

4. In the introduction to pBMA in the Results section, a brief mention of the ELPD formula, explaining that higher values indicate better model performance, would help readers interpret the model comparison in the subsequent figures.

We agree, and now define and discuss the use of the ELPD on lines 126–129.

Reviewer #2:

The manuscript "Increasing certainty in systems biology models using Bayesian multimodel inference" by Linden-Santangeli et al. presents a framework for performing inference using "multimodel inference" (MMI). The authors present a brief review of MMI, and through a case study of ERK signaling, demonstrate the advantages of MMI over alternatives.

The manuscript is very well written, and likely a useful contribution. My biggest criticism, however, is that it is unclear from the outset how the framework that the authors develop is different from existing MMI frameworks. Specifically, it is difficult to tell whether the paper should be considered a review of MMI, which is already well developed, that focusses on one case study, or whether the authors have significantly expanded upon existing MMI frameworks and are presenting a new statistical method.

We thank the reviewer for thoroughly assessing our work. Please see the discussion on the novelty of the work under the note to the editor at the beginning of this document.

I also have a few other comments and concerns, that I believe should be addressed before the paper could be considered for publication.

1. Choice of model for a case study. I agree that ERK is potentially a good case study, as there are often many potential candidate models that can describe any amount of data equally well. However, a drawback (of either the choice of system, or the implementation) is that all the models can be considered to be nested inside a larger model. Thus, the pure Bayesian approach to MMI would be equivalent to performing inference on this larger model, subject to a spike-slab type prior which restricted model parameters such that only the chosen k models can arise. The authors don't appear to acknowledge this, and I can't see why, if *prediction uncertainty* is the main goal, a MMI-based approach would be better than simply performing inference on the "super-model". I push back on the authors suggestion that MMI is critical to select the best mechanism for the data as it avoids making a priori assumptions: by its very nature, MMI effectively does make very strong a priori assumptions (albeit, perhaps less so than many studies that utilise a single model).

We appreciate these thoughtful comments. We agree that the ERK models we study have related network structures but vary in complexity, with some appearing to be simplified versions of others. However, we disagree that the models can be nested inside a larger model and then recovered using the spike-and-slab prior approach suggested by the reviewer. If this was the case, then we should be able to derive the individual models by eliminating reactions and pathways from the larger model. This is not necessarily possible. For example, consider two of the models that we analyze. The Kholodenko 2000 model [1] includes the core ERK signaling cascade and a single ERK-to-Sos feedback loop, while the Kochanczyk 2017 model [2] additionally includes intermediate species and more feedback loops (see Figure 2 in the revised manuscript). Furthermore, the Kholodenko 2000 model uses Mass-Action kinetics, while the Kochanczyk 2017 model uses a mixture of Michaelis-Menten and Hill-type kinetics. Although the network structures of these models may be nested, it is not clear how they would be derived by eliminating components from a larger model because they each make different approximations to capture similar sets of reactions. Thus, as we understand it, the author's suggested approach of using sparsifying spike-slab type priors on a larger would not be able to capture the sub-models.

However, we do agree that Bayesian sparse identification approaches (such as the approach mentioned by the reviewer) do have the ability to generate new candidate models that can be useful for model selection and be assessed using Bayesian MMI. For example, a similar approach was developed by Mangan et al.[3] in the frequentist setting. We had previously touched on this in the Discussion section, but we now added a new paragraph that emphasizes alternatives to Bayesian MMI (lines 453–462).

Additionally, we have reworded the final results subsection to remove language such as "MMI was critical" (see, e.g., lines 400–401) and to better contextualize the results.

2. On the above comment, it is perhaps, then, no surprised that the MMI approach performs well for the case study considered, as all models can fit the data well. It would be a useful (perhaps even critical) comparison if the authors included several models that were not able to capture the data.

We agree that including models that do not fit the data well is important to this study. In the example with synthetic dose-response data in Supplemental Figure S4 several models do not fit the data well. From these comparisons, we can see that MMI weights models that fit the data best (as measured by the ELPD or marginal likelihood) and does not weight models that do not fit the data well. Thus, we show that MMI predictions are robust to *bad* models. We updated the results on lines 212–215 to highlight this point.

3. In the hypothetical limit than an infinity of models (or a large amount) are used, is it possible that the MMI approach would behave similarly to a Gaussian-process-like (or purely machine learning-like) method?

While there may be a hypothetical limit of infinite models, for the well-studied ERK pathway, there is a limit of a large number of models available (around 100 or so). This is similar to other well-studied pathways, such as NF κ B. Thus, we aim to characterize MMI in the practical setting where a finite number

of models are available.

On the other hand, mathematically, this is an interesting point. However, we do not believe that we should expect MMI to behave like a data-driven machine learning approach in the infinite model limit. A deeper mathematical analysis would be required to understand this limit, which we haven't done for the above reasons. We refer the reviewer to Hoeting *et al* 1999 [4] (and the references within) for an in-depth review of MMI methods with larger model sets.

4. The authors focus throughout on "reducing uncertainty". I am not convinced that this is the best metric to assess the suitability of a statistical method. A more convincing goal would be to establish the approach that most accurately quantifies uncertainty. For example, the "best framework", by the authors current methodology, could be one that predicted very little variance in model predictions. However, this would never be expected from a finite amount of experimental data. Thus (following from comment (1), above), I am not yet convinced (but could be) from the authors present work that alternatives to Bayesian-MMA, which has a very clear statistical justification, perform better. To test this, I would imagine that one would need to predict outside the data range (i.e., different inputs) and validate on unseen experimental (or synthetic) data.

We appreciate the thoughtful suggestions. First, we originally chose to assess uncertainty of posterior densities (and focus on posterior uncertainty reduction), because when used to estimate posterior predictive densities, which combine posterior uncertainty with uncertainty in the data, lower posterior uncertainties equate to the data uncertainty dominating over uncertainty due to parameter estimates. We agree that the original presentation did not acknowledge this well, so we now discuss this point in a new paragraph in the Discussion (lines 424–430). We also removed some language around reducing uncertainty to focus more on accurately representing uncertainty.

Second, in the examples in Figure 5 and S8, we compare the MMI methods using predictions outside of the training data range (predictions of ERK activity at times beyond what is used for training) and do show that MMI predictions have bounded uncertainty intervals that better align with the data and have lower predictive error. We now make sure to highlight that these predictions are of unseen data in the results section (lines 294–296) and Discussion (lines 429–430)

5. (minor) The authors should be more careful in their use of statistically loaded words like "accurate and certain predictions" (this could, potentially, have confused my reading the the paper and led to comment (4) above).

We have revised the manuscript to limit our usage of these words to cases when we directly discuss data.

Reviewer #3:

In this paper, the authors employ Bayesian multi-model inference (MMI) to establish and test a framework to reduce uncertainties and improve the predictive capabilities of a suite of models employed in systems biology. Specifically, they employ ten models of extracellular-regulated kinase (ERK) signaling to pose and illustrate the framework. However, the framework is very generally and will equally apply to a range of deterministic models employed to predict other phenomena in systems biology. They focused on two primary questions: (i) what techniques can be employed to reduce uncertainty on model predictions, and (ii) what are techniques to reduce model choice uncertainty to improve the certainty of model predictions.

To address these issues, they cite previous work in which they employed sensitivity analysis to determine model parameters which significantly influence considered model responses. They subsequently applied Bayesian inference to calibrate parameters in the ten models using training data. The primary novelty is the main step in which they employed Bayesian multi-model inference to combine specified models to reduce uncertainty and improve the resulting model predictions. For this final step, they employed three methods for choosing the weights: Bayesian model averaging (BMA), pseudo-Bayesian model averaging (pseudo-BMA), and stacking of predictive densities (stacking). Using this framework, they demonstrated the capability of significantly decreasing uncertainty and improving model predictions in the context of the ten ERK models. In the process,

they address the following broad questions:

- Is there a best Bayesian multi-model inference methodology for systems biology?
- How much data are needed for effective multi-model inference?

Overall, this is a very strong paper and use of Bayesian multi-model inference constitutes an important framework for improving the accuracy of models employed for systems biology. Whereas demonstrated in this context, the use of Bayesian multi-model inference is equally important in other biological, physical, and engineering applications employing deterministic predictive models; e.g., ODE or PDE models. Moreover, the authors very carefully detail and highlight both the implementation of the framework and its ramification for the ERK models. These constitute significant strengths of the paper and augment its impact in the field.

We thank the reviewer for their detailed review and for highlighting the significant strength of our work.

There is one substantive comment and several more minor comments which should be addressed before the paper is accepted for publication.

Substantive Comment

This comment pertains more to the presentation style than the content. The authors are correct that the primary contribution of the paper is the use of Bayesian multi-model inference (MMI) in the context of systems biology rather than development a fundamental statistical framework for Bayesian MMI. To establish this contribution, they very carefully detail the advantages of the framework with the three techniques for choosing modeling weights in the context of the ten ERK models. In this sense, the paper is very strong. The potential issue is that this attention to detail for the specific models may reduce the impact of the paper for readers in other areas of systems biology or science. For example, the discussion on page 12 very clearly establishes the impact of the MMI techniques for the ERK models but it may be difficult to follow for readers from other research areas. My recommendation to address this would be to slightly refine some of the discussion to further highlight high-level contributions of the framework in the context of general systems biology models, as indicated in the title. This might be achieved by rewording some of the discussion or further highlighting key concepts.

We appreciate the helpful suggestions. Following the reviewer's advice, we highlighted the general applicability of MMI in the caption of Figure 1 and on lines 112–113. Additionally, we revised the Discussion section to place the results in a more general systems biology context and to highlight potential impacts beyond ERK signaling. For example, see lines 417–420.

Minor Comments

1. The authors cite previous work in which they used sensitivity analysis to determine influential parameters for the analysis in this paper. It would be good to include a sentence indicating the number of original model parameters and the reduced number employed in this analysis. This would highlight the importance of this step to readers.

Thank you for the good suggestion. We now reference Supplemental Table 1, which states the number of original model parameters and reduced numbers for each model on lines 173–174.

2. The authors detail on page 15 the manner in which they generate unbiased Gaussian observation errors. It would good to indicate if the errors are additionally independent.

Yes, the errors are assumed to be independent. We now clarify this on line 536.

3. On page 19, it was stated that they used a 4/5 implicit Runge-Kutta method to numerically integrate the ODE systems. It would be advantageous to very briefly discuss whether these systems are stiff and, if so, why they did not employ a stiff ODE routine to improve efficiency.

Most of the ODE systems that we solve are potentially stiff systems. We now clarify on lines 686–693 that the ODE solver that we used, the Kvearno 4/5 implicit Runge-Kutta scheme [5], is well suited for stiff

systems and that the PID-controller-based adaptive time-stepping scheme [6–8] also aids in integrating stiff systems.

4. There seemed to be some confusion regarding the nature of reported uncertainty intervals for responses. In the Discussion section on pages 16-17, they noted the use of posterior push-forward intervals for the Quantity of Interest (QoI) when sampling parameter uncertainty and posterior predictive distributions when sampling both parameter and data uncertainty. However, in earlier discussion and figure captions, they appear to refer to the former as credible intervals, which is more common when referring to intervals in the QoI due to parameter uncertainties. The consistency of these terms should be checked.

Thank you for highlighting this, and we agree that it was a point of confusion in our previous version of the manuscript. Throughout the paper, we use *95 % credible intervals* to display and quantify uncertainty; however, we had previously computed credible intervals of both posterior predictive densities when evaluating within-sample predictions and posterior (or posterior push-forward) densities when evaluating out-of-sample predictions. We now only consider posterior densities to avoid confusion and because they better highlight model-to-model differences in predictions. In doing so, we updated Figure 6 (previously Figure 5) and Supplemental Figure S4 to now show posterior densities instead of posterior predictive densities. We note that posterior densities show *lower uncertainty* because they do not account for uncertainty in the data, only uncertainty due to model parameters. Also, see our response to Reviewer 2, comment 4. We have removed the aforementioned notes from the discussion.

5. The notation employed for citing other works was confusing and somewhat nonstandard; e.g., H' 1996 for Huang and Ferrell 1996. Unless this is required by the journal, it would probably aid readers to include the more complete references.

We appreciate the reviewer's suggestion to change the notation for citing models. However, we chose the shortened notation to avoid excess clutter in the figures and continue that through the text for consistency. We have double-checked that all usage remains consistent throughout the manuscript. We note that all other references to other works follow standard notation.

6. It is common to additionally perform Bayesian model averaging using stochastic search variable selection (SVSS). This does not need to be discussed but it would aid readers to include a citation; e.g., B.J. Reich and S.K. Ghosh, Bayesian Statistical Methods, CDC Press, 2019.

Thank you for the suggestion. We now include a reference to SVSS in the discussion section on lines 457.

Note that these comments are all minor and can be easily addressed with minimal additional discussion.

In summary, the use of Bayesian multi-model inference to develop a framework to improve the accuracy and predictive capabilities of systems biology models is timely and important. Moreover, the authors demonstrate that the considered framework isolates and reduces uncertainties pertaining to the suite of considered models. Finally, they demonstrate that the resulting model predictions are robust with regard to uncertainty in experimental and training data. These constitute significant strengths of the paper. However, the noted issues should be addressed before the paper is considered for publication in Nature Communications.

Reviewer #4:

The authors have conducted a complex study of multi-model inference on biological cellular signaling data, comparing multiple existing weighting methods. The paper highlights the challenges of combining models to fit to biological signaling data. Overall, they show that multi-model inference increases dynamic predictions and decreases uncertainty on those predictions, primarily on in-sample data. They then use their framework to evaluate a small subset of ERK signaling pathway models showing that models built to be consistent with existing hypothesis can fit the data. Overall, I think the paper is a sound contribution to the field of mathematical modeling and inference in biological systems. I believe the overall approach of multi-model inference is a promising and necessary direction. However, I had some issues with the strength of the claims made by the authors and believe either some re-wording or additional analysis is needed to strengthen the paper and ensure the evidence supports the claims.

Major Revisions:

1. More clarity on what is novel. Specifically, the authors use language indicating that the framework is novel, but it's unclear which components are new. Is this a methods comparison paper? A new methods paper? Are the methods for propagating uncertainty new? A biology paper? It seems that it is mostly a methods comparison/demonstration paper, showing how multiple existing methods can be combined to analyze complex biological problems. It is not clear what biological insight is new (see last revision suggestion).

Please see our comments to the editor clarifying the novelty of this work at the beginning of this document.

2. What is the connection/expectations of predictions when training on one data-set vs another? Are different parameters estimated? How different are the models across data sets? It seems that very little out of sample prediction was done (only some projection in Figure 4). Is it possible to train on one type of data and predict on another? Or are the underlying fit models/mechanisms expected to be different between data-sets? If so, what is the potential for predictive modeling using MMI? Can it only be used to infer parameters and try to draw conclusions on each individual dataset?

The difference between training on one dataset versus another is that different posterior densities are learned because they are conditioned on different data sets. For example, Supplemental Figure S9 compares the marginal posterior cumulative density functions for models trained on the cytoplasm data (Figure 3, previously Figure 2) to models trained on the plasma membrane data (Supplemental Figure S3).

It is possible to train on one type of data and predict another, for example, training on dynamics and predicting dose responses. However, the reviewer is correct that the underlying posterior estimates would be different between types of data. The following figure (*NOT currently included in the manuscript*) shows an example where we train the models on trajectory data and predict dose responses. Here, we see that the models that capture the trajectory data the best (highest ELPD) are not necessarily those that predict the dose-response curves best. Therefore, the resulting MMI predictions of the dose-response curve reflect the performance of the models that receive weights based on how well they predict trajectory data. We found that it is even more challenging to predict trajectories from models trained on dose-response data because the resulting parameter estimates were not restricted to capturing the timescales of activation, simply the strength of activation. We note that these are limitations of fitting models to data and are not limitations of MMI, so we chose not to include these results in the manuscript. However, if the reviewer feels that this figure is a valuable addition to the paper, we are glad to include it in the supplement.

Further, the reviewer is correct that Figure 5 (previously Figure 4) contains out-of-sample forecasts of ERK activity. We believe such forecasting is a realistic case where out-of-sample predictions would be used in practice. We have revised the associated text to better highlight that Figure 5 shows out-of-sample predictions.

Out-of-sample predictions of the EGF-ERK dose-response curve with models trained on trajectory data. (A) Synthetic trajectory data was generated by simulating the H' 1996 mode for 30 minutes at three EGF input levels (0.001, 0.005, and 0.106 nM). (B) Posterior densities of trajectories for a subset of models ordered by ELPD. (C) Posterior densities of predicted dose-response curves using models trained on trajectory data. (D) ELPD of models. (E) Model probabilities. (F) Model weights. Note that D-F were calculated based on fits to trajectory training data. (G) RMSE of posterior dose-response predictions. (H) Uncertainty of posterior dose-response predictions.

3. I would like to see more insight/analysis of structural/mechanistic details in models. The authors start out with the question : “(1) How can we quantify the effects of uncertainty in the model formulation, called model uncertainty, on model predictions?” but do not provide enough detail of the variation in model formulation or it’s effects on predictions to answer this question. The lack of visualization makes much of the text around the different models and signaling pathway structure difficult to follow. It would be helpful to have some schematic of the ERK signaling pathway which visually displays differences in different models in terms of their mechanistic complexity (at some course-grained level). Why were specific model selected or weighted more heavily? Is that because they are more complex? Or contain necessary mechanisms? The authors note they selected the models based on the idea that they contain different mechanisms, but do not follow up on why specific models may be weighted more heavily for one type of data-set or another. Generally, the presentation across models are difficult to follow because they are either high level (showing uncertainty on plots), complex (list of parameter values in the supplement), or difficult to connect with a diagram of mechanism (description in the text).

Thank you for the helpful suggestions and insightful questions. We have made several changes to clarify the discussion of the models:

- (a) We added Figure 2, which depicts the ERK signaling network at a high level and denotes which models contain which of the components.
 - (b) We revised the introduction of the models to focus on these components on lines 152–160.
 - (c) We added several sentences in the first results section reflecting on the mechanistic differences between the best and worst models on lines 194–199. We attempted to limit this discussion of mechanistic detail to improve readability and ensure that the work appeals to a wide audience (See Reviewer 3 “substantive comment”).
4. In a few places in the text the authors discuss model selection vs model weighting. In some cases, model selection is presented as undesirable (pg 4 Line 116), but in others (such as the discussion of localization) model selection is the goal. Can the authors clarify when one would want to select a specific model vs train a weighted ensemble? And what the goals for each section of their paper is? In particular, please reconcile the idea that stacking produces undesirable results with the over all goal of selecting the most likely mechanisms.

Thank you for the helpful suggestions and questions. We added text to clarify whether *model averaging* or *selection* is the goal of each section (see, e.g., lines 177-178). Additionally, we revised the discussion section to delineate between the two use cases and compare the methods in both (see, e.g., lines 410–411). In light of new results (Figure 6 and Supplemental Table 4), which show stacking not only selecting a single model, we no longer emphasize that point.

5. The logic of the localized ERK activity dependence seems circular. Only parameters associated with Rap1 and negative feedback were allowed to vary between locations. These parameters were chosen because of previous findings in the literature. The authors showed that models with only these parameters varying could be fit to the data, but they do not test whether any other mechanisms could account for the differences between cytoplasmic and plasma membrane ERK activity. To do this, the authors would need to show that no other set of parameters could reproduce the behavior. While this shows that the MMI can find a consistent fit to existing hypothesis it does not show “... MMI would select the models that best reflect the hypotheses that agree with the available data.” (pg 12 line 347-348), “Location-specific Rap1 signaling and ERK negative feedback were necessary to predict ERK activity’s sub-cellular variability.” (pg 12 Line 349- 350), or “Thus, these results show that ERK negative feedback and Rap1 activity are necessary for location-specific ERK activity... These findings suggest that cytoplasmic ERK activity depends on ERK negative feedback, while plasma membrane ERK activity depends on Rap1. Based on the MMI-selected hypothesis, we propose a new ERK negative feedback- and Rap1-dependent model for sub-cellular variability in ERK activity (Figure 5F). Here, MMI was critical to selecting the best mechanism from the available data without making any a priori assumptions that a single hypothesis

would be best.” (pg 12 line 385 – 391).

My main complaint here is that a small subset of mechanisms were pre-selected based on existing literature. It is not clear what the authors mean that “MMI-selected” hypothesis. The MMI selects between models but it does not select between the feedback, Rap1 activity and all other changes that could be made. Interestingly the authors note that the initial models (Fig 2B), allowed all parameters to vary between cytoplasm and plasma membrane. However, they merely claim that allowing all parameters to vary between compartments is not biologically feasible rather than analyzing whether the variance in the parameters when all are allowed to be free suggests either the same mechanisms identified in the literature or alternatives. It would be interesting to know if the less restricted models suggest something new. More generally, the authors need to clarify what was learned specifically from MMI vs from pre-selection based on previous literature and be precise in their conclusions.

Thank you for the thoughtful comments and suggestions. We originally focused specifically on Rap1 and ERK negative feedback to ensure that our analysis of subcellular location-specific ERK signaling remained grounded in biological reality. We also worked closely with our co-author, Jin Zhang, whose lab specializes in sub-cellular signaling, to choose these. However, we agree with the reviewer that our original presentation of these results did not include enough context and that there is value in including additional analyses. We made the following additions and changes to the manuscript:

- (a) Following the reviewer’s suggestion, we conducted a computational study allowing all possible combinations of free parameters to vary between subcellular compartments. In doing so, we interestingly found that there are, in fact, many possible combinations of parameters that can fit the data (Supplemental Table 4). However, the compartment-specific parameter estimates selected by MMI (Supplemental Table 4) do not align with the current, albeit still evolving, knowledge of sub-cellular ERK signaling. We added a paragraph on lines 349–359 of the results to discuss these new findings.
- (b) We motivate the study presented in Figure 6 (previously Figure 5), which focuses on Rap1 and ERK negative feedback as *comparing hypotheses* about what drives sub-cellular differences in ERK signaling, rather than *selecting the mechanism* that drives the data. We additionally ensure that the discussion of these results highlights that the findings were based on the pre-selected hypotheses rather than being selected by MMI. Importantly, we continue to highlight how MMI enables the comparison between the pre-selected hypotheses.

Minor comments:

Unclear what is meant by “comparable model representations” line 145 pg 4

We have removed “comparable model representations” and revised the sentence. It now reads: *Even though each model was initially used to answer different research questions, we chose these ten because each one represents the ERK signaling pathway from input to ERK activation.* (lines 151–153)

Figure S5D why does excluding the worst fit model seem to increase the RMSE?

In Figure S5D (along with Figure 4D and S6D), excluding the worst fit model has very little impact on the RMSE compared to the MMI estimates with all models, which is expected. For clarification, the bars with dotted patterns show the cases in which the worst models are excluded.

References:

1. Kholodenko, B. N. Negative feedback and ultrasensitivity can bring about oscillations in the mitogen-activated protein kinase cascades. en. *Eur. J. Biochem.* **267**, 1583–1588 (Mar. 2000).
2. Kočańczyk, M. *et al.* Relaxation oscillations and hierarchy of feedbacks in MAPK signaling. en. *Sci. Rep.* **7**, 38244 (Jan. 2017).

3. Mangan, N. M. *et al.* Inferring Biological Networks by Sparse Identification of Nonlinear Dynamics. *IEEE Transactions on Molecular, Biological and Multi-Scale Communications* **2**, 52–63. ISSN: 2372-2061, 2332-7804. (2024) (June 2016).
4. Hoeting, J. A. *et al.* Bayesian model averaging: a tutorial. en. *SSO Schweiz. Monatsschr. Zahnheilkd.* **14**, 382–417 (Nov. 1999).
5. Kværnø, A. Singly diagonally implicit Runge–Kutta methods with an explicit first stage. *BIT Numerical Mathematics* **44**, 489–502 (2004).
6. Söderlind, G. Digital Filters in Adaptive Time-Stepping. *ACM Transactions on Mathematical Software* **20**, 1–26 (2003).
7. Söderlind, G. Automatic control and adaptive time-stepping. *Numerical Algorithms* **31**, 281–310 (2002).
8. Hairer, E. *et al.* *Solving Ordinary Differential Equations II Stiff and Differential-Algebraic Problems* Second Revised Edition (Springer, Berlin, 2002).

Best regards,

Padmini Rangamani, on behalf of all the authors

DEPARTMENT OF PHARMACOLOGY
UCSD SCHOOL OF MEDICINE

9500 GILMAN DRIVE
LA JOLLA, CALIFORNIA 92093-0404
FAX: (858) 534-7029

PADMINI RANGAMANI
PROFESSOR, PHARMACOLOGY
PROFESSOR, MECHANICAL AND AEROSPACE ENGINEERING
UNIVERSITY OF CALIFORNIA, SAN DIEGO
CELLULAR AND MOLECULAR MEDICINE EAST, ROOM 2050
prangamani@health.ucsd.edu (858) 534-4046

July 11, 2025

Ref: Manuscript NCOMMS-24-69628 (“Increasing certainty in systems biology models using Bayesian multimodel inference”)

Dear Reviewers,

We would like to thank the reviewers for providing insightful feedback and suggestions throughout this review process and for their support of our work. In what follows, we provide a point-by-point response to the individual comments made by the reviewers. Original comments are reproduced in black and our response is shown in blue.

Point-by-point response:

Reviewer #2:

(Remarks to the Author)

I thank the authors for addressing my original concerns. I have no further comments to add.

(Remarks on code availability):

The code is well commented and easy to download. I have not reviewed or ran the code as I do not use Python.

Thank you for your support of our work.

Reviewer #3 (Remarks to the Author):

In this paper, the authors employ Bayesian multi-model inference (MMI) to establish and test a framework to reduce uncertainties and improve the predictive capabilities of a suite of models employed in systems biology. Specifically, they employ ten models of extracellular-regulated kinase (ERK) signaling to pose and illustrate the framework. However, the framework is very generally and will equally apply to a range of deterministic models employed to predict other phenomena in systems biology. They focused on two primary questions: (i) what techniques can be employed to reduce uncertainty on model predictions, and (ii) what are techniques to reduce model choice uncertainty to improve the certainty of model predictions.

To address these issues, they cite previous work in which they employed sensitivity analysis to determine model parameters which significantly influence specified model responses. They subsequently applied Bayesian inference to calibrate parameters in the ten models using training data. The primary novelty was the main step in which they employed Bayesian multi-model inference to combine specified models to reduce uncertainty and improve the resulting model predictions. For this final step, they employed three methods to choose the

weights: Bayesian model averaging (BMA), pseudo-Bayesian model averaging (pseudo-BMA), and stacking of predictive densities (stacking). Using this framework, they demonstrated the capability of significantly decreasing uncertainty and improving model predictions in the context of the ten ERK models. In the process, they address the following broad questions: • Is there a best Bayesian multi-model inference methodology for systems biology? • How much data are needed for effective multi-model inference?

Overall, this is a very strong paper and use of Bayesian multi-model inference constitutes an important framework for improving the accuracy of models employed for systems biology. Whereas demonstrated for models in systems biology, the use of Bayesian multi-model inference is equally important in other biological, physical, and engineering applications employing deterministic predictive models. Moreover, the authors very carefully detail and highlight both the implementation of the framework and its ramification for the ERK models. These constitute significant strengths of the paper and augment its impact in the field. Finally, the authors have adequately addressed the issues which I detailed for the original submission.

In summary, the use of Bayesian multi-model inference to construct a framework to improve the accuracy and predictive capabilities of systems biology models is timely and important. Moreover, the authors demonstrate that the considered framework isolates and reduces uncertainties pertaining to the suite of considered models. Finally, they demonstrate that the resulting model predictions are robust with regard to uncertainty in experimental and training data. I now recommend that the paper be accepted for publication in Nature Communications.

We appreciate the positive feedback and support of our work.

Reviewer #4 (Remarks to the Author):

Thank you to the authors for their extensive consideration of my comments. I am impressed with the exercise they undertook to explore additional models and clarify whether multiple mechanisms can fit the data.

I encourage the authors to include the out-of-sample prediction figure in the supplement. I do not think it is necessary, but I find it satisfying that some models do very well. They appear to have low confidence, but the predictions of the dose response curve are within the error bars for the "better models," which I think is quite impressive. I would have been troubled if they were all wrong or if the confidence intervals were overly tight for this type of out-of-sample test. That said, if the more realistic usage is projected forward in time, I understand why they may not want to include unrealistic usage.

I support the publication of the manuscript.

Thank you for the support of our manuscript. While we agree that the out-of-sample results shown in the previous response document are satisfying, we have chosen to exclude them from the final manuscript. We do not think that they are necessary to the overall story and do not wish to add additional distractions.

Best regards,

Padmini Rangamani, on behalf of all the authors